# Graded decisions in the human brain

Tao Xie [1,2], Markus Adamek [1,2], Hohyun Cho [1,2], Matthew A. Adamo [3], Anthony L. Ritaccio [4,5], Jon T. Willie [1,2], Peter Brunner [1,2,4] ✉ & Jan Kubanek [6] ✉

Decision-makers objectively commit to a definitive choice, yet at the subjective level, human decisions appear to be associated with a degree of uncertainty. Whether decisions are definitive (i.e., concluding in all-or-none choices), or whether the underlying representations are graded, remains unclear. To answer this question, we recorded intracranial neural signals directly from the brain while human subjects made perceptual decisions. The recordings revealed that broadband gamma activity reflecting each individual's decision-making process, ramped up gradually while being graded by the accumulated decision evidence. Crucially, this grading effect persisted throughout the decision process without ever reaching a definite bound at the time of choice. This effect was most prominent in the parietal cortex, a brain region traditionally implicated in decision-making. These results provide neural evidence for a graded decision process in humans and an analog framework for flexible choice behavior.

Many cognitive processes, including decision-making, involve deliberation over a brief period of time. Psychology, neuroscience, and neuroeconomics have debated over how the process of deliberation is implemented at the neural level. It has been found that many brain regions track the evidence accumulated for a decision[1–13], which has been captured by a dominant, drift-diffusion model of decision-making[14–16].

However, it remains unclear how the accumulation process concludes. The traditional view posits that choices are made in an all-or-none manner[14–25], when a neural signal that represents a forming decision reaches a fixed bound (Fig. 1a). More recently, this view of the decision process has been criticized. In particular, it has been questioned whether the concept of a decision bound provides an inclusive-enough account of behavioral and neural data[26–33]. For instance, humans and animals generally make decisions under time constraints, which exert limits on the available decision time[1,3,6,34–39]. These constraints have been modeled by collapsing bounds that decrease their levels over the decision time[28,40]. Furthermore, there are alternative, multidimensional attractor network models that do not require a definition of a bound[41–44]. In attractor network models, decision-related activity evolves in a multidimensional and hierarchical space involving many brain regions and neurons until it reaches a stable state defined by time constraints, accuracy requirements, and other decision-relevant variables. Both the collapsing and attractor network models allow for decision-related activity to be graded by evidence accumulated at the time of choice (Fig. 1b).

The two prevalent models (Fig. 1) have been difficult to tease apart. Computational models often make similar predictions of choice behavior even though they invoke fundamentally different neural mechanisms[25,28]. The models could be distinguished using direct recordings of neural activity, but such recordings have thus far only been conducted in particular brain regions of non-human primates[1–5]. Since specific brain regions encode different aspects of forming decisions, a conclusive answer to how the brain represents the entirety of the decision process in humans has remained elusive. On the other hand, human studies, which have used non-invasive modalities[6–13], could only access broadly distributed low-frequency signals, which provides a coarse perspective on the underlying neural processes.

To address this issue, we have recorded local field potentials (LFPs) directly from multiple regions of the human brain during perceptual decisions (Figs. 2 and 3). The broad coverage of the intracranial

[1]Department of Neurological Surgery, Washington University School of Medicine, St. Louis, MO 63110, USA. [2]National Center for Adaptive Neurotechnologies, St. Louis, MO 63110, USA. [3]Department of Neurosurgery, Albany Medical College, Albany, NY 12208, USA. [4]Department of Neurology, Albany Medical College, Albany, NY 12208, USA. [5]Department of Neurology, Mayo Clinic, Jacksonville, FL 32224, USA. [6]Department of Biomedical Engineering, University of Utah, Salt Lake City, UT 84112, USA. ✉e-mail: pbrunner@wustl.edu; jan.kubanek@utah.edu

recordings and their high fidelity enabled us to characterize the neural dynamics and the brain regions involved in decision formation.

## Results

### Decision tasks and behavior

We recorded the neural dynamics underlying deliberation using intracranial electrodes (Fig. S1) while eight naive human subjects assessed temporally defined quanta of decision evidence (Fig. 2). The evidence quanta constituted Poisson-distributed click sounds delivered into the right and left ears[45]. The subjects communicated their choices using movements of two distinct choice kinds (a saccade or a button press) in two task contexts (congruent or reversed). In the congruent task context (Fig. 2b-top), subjects implanted with electrode grids over the left hemisphere made a saccade (SC) choice to a left target if they heard more click sounds delivered into the left ear, and made a button press (BP) choice with the right hand if they heard more click sounds delivered into the right ear. This contingency was flipped in the reversed task context (Fig. 2b-bottom). In both choice kinds, subjects were free to indicate their choice following the

stimulus onset by making the respective movement. The stimulus ceased upon a choice. A total of 13 sessions were recorded in the two task contexts (Fig. S1).

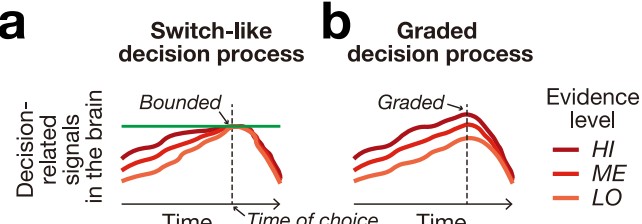

**Fig. 1 | Nature of developing decisions. a** One class of models posits that choices are made when brain activity reaches a fixed bound, which results in all-or-none, switch-like choices. **b** An alternative view is that the choice process is generally graded. The dashed vertical line represents the time of choice. The LO (low), ME (medium), and HI (high) levels indicate the amount of evidence for a choice.

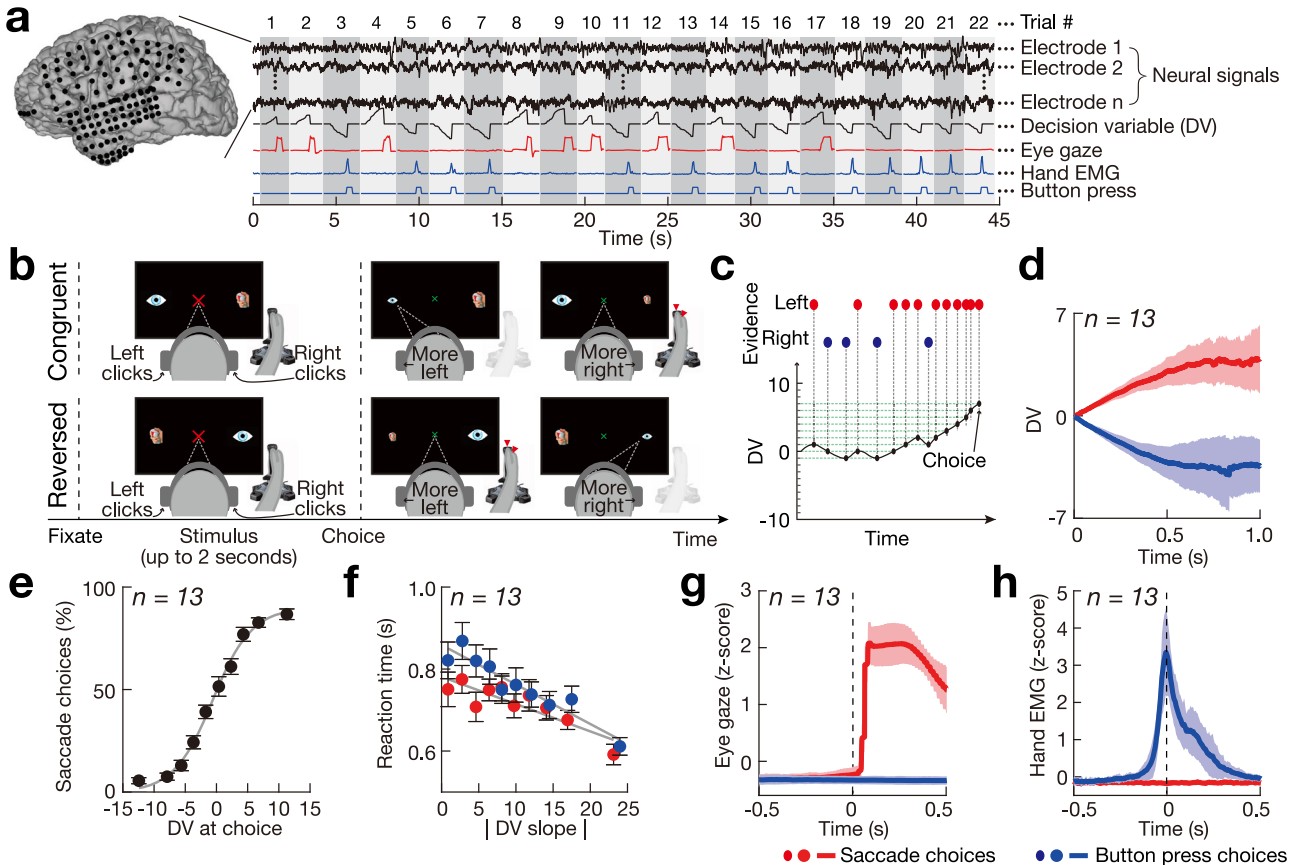

**Fig. 2 | Neural recordings, tasks, and behavior. a** We recorded the intracranial activity of the human cortex during fixation, saccadic, and manual responses. **b** After acquiring a fixation cross, subjects listened to a binaurally presented auditory stimulus. Subjects decided whether they heard more click sounds in the left or right ear. In the congruent task context, subjects (with electrodes implanted in the left hemisphere) made a saccade (SC) to the left side if they heard more clicks on the left, and made a button press (BP) if they heard more clicks on the right. This sensorimotor contingency was flipped in the reversed task context. Subjects were free to indicate their choice following the stimulus onset by making the respective movement. The auditory stimulus ceased upon a choice. A total of 13 sessions were recorded in 8 subjects in the two task contexts (n = 13). Specifically, 7 subjects performed the congruent task, 6 subjects performed the reversed task, and 5 subjects performed both tasks. **c** Decision variable (DV) computation during an example trial. Red/blue dots indicate auditory clicks in the left/right ears. **d** DV

(mean ± s.d., n = 13) as a function of time, separately for trials that resulted in a SC (red) and BP (blue) choices. One second following the stimulus onset, the DV reached 3.9 ± 2.4 and −3.4 ± 1.8 (mean ± s.d., n = 13) for SC and BP choices, respectively. **e** Proportion (mean ± s.e.m., n = 13) of SC choices as a function of the DV at the time of choice. The psychometric curve fitting the data of each subject explained 92.4 ± 3.2 % of the variance in the choice behavior (mean ± s.d., n = 13). **f** Reaction time (mean ± s.e.m., n = 13) as a function of the absolute value of the DV slope for SC (red) and BP (blue) choices. The slope of this relationship was −7.4 ± 6.8 and −10.5 ± 8.5 (mean ± s.d., n = 13) for SC and BP choices, respectively. **g, h** Eye gaze and hand EMG signals (mean ± s.d.) for trials that resulted in a SC (red) and BP (blue) choices, respectively. Around the time of the choice, the separation, measured by Cohen's d, was 10.5 ± 3.3 and 3.0 ± 0.7 (mean ± s.d., n = 13) for the eye gaze and EMG signals, respectively.

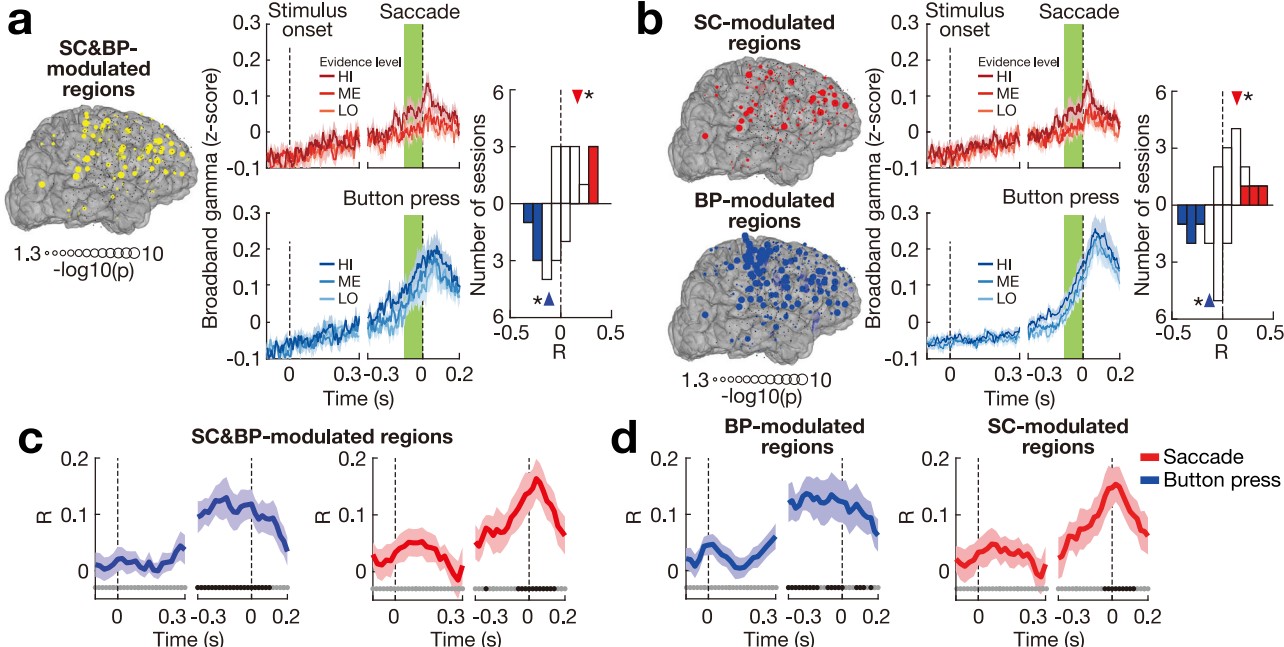

**Fig. 3 | Effector-related broadband gamma (γ) signals index developing decisions and remain graded during choice. a** Decision signals in SC&BP-modulated regions. Left panel: The yellow symbols indicate SC&BP-modulated electrodes that show significant ($p < 0.05$, corrected using false discovery rate, one-tailed randomization tests) broadband gamma (γ) modulation during saccade (SC) and button press (BP) choices compared with baseline. Middle panel: Session-mean (±s.e.m., $n = 13$) γ activity over all SC&BP-modulated electrodes. The signals are aligned to stimulus onset (left dashed line) and movement onset (right dashed line). The top (bottom) panel shows γ activity for trials that resulted in a SC (BP) choice. Right panel: Spearman's correlation $R$ between trial-by-trial values of DV and γ activity around the time of choice (green bars). The individual $R$ values in the histogram represent the correlation for individual sessions ($n = 13$) and are presented separately for SC (top histogram) and BP (bottom histogram) choices. The color-filled bars denote significant $R$ values ($p < 0.01$, one-tailed randomization tests). The triangle denotes the average $R$. *$p = 7.3 \times 10^{-3}$ and $p = 1.5 \times 10^{-3}$ for SC and BP

choices, respectively (two-tailed t-tests). **b** Decision signals in SC/BP-modulated regions. Left panel: The red and blue symbols indicate electrodes that show significant ($p < 0.05$, corrected using false discovery rate, one-tailed randomization tests) γ modulation during SC and BP choices compared with baseline, respectively. Middle-right panels: Same analyses and format as (**a**), separately for the SC-modulated and BP-modulated electrodes. *$p = 7.5 \times 10^{-3}$ and $p = 4.7 \times 10^{-3}$ for SC and BP, respectively (two-tailed t-tests). **c, d** Session-averaged (±s.e.m., $n = 13$) correlation between the γ activity and DV at the time of choice as a function of time, aligned in the same way as in (**a, b**). The blue and red lines show the average $R$ for BP and SC choices, respectively. The black dots near the time axis mark the times during which the correlation was significant ($p < 0.01$, $n = 13$, one-sample two-tailed t-tests), while the gray dots indicate no significant correlation ($p > 0.01$, $n = 13$, one-sample two-tailed t-tests). **c, d** provide data for decision signals in the SC&BP-modulated and BP/SC-modulated regions, respectively.

The temporally defined quanta of evidence and each subject's choices provided a decision variable (DV; Fig. 2c) that captured the subjects' decisions, as in previous studies[15,46]. The DV faithfully explained the subjects' choices across the sessions. First, as expected, the polarity of the DV diverged over time according to each subject's choices (Fig. 2d). Crucially, the DV at the time of choice captured the probability of choosing either alternative (Fig. 2e). Moreover, in line with chronometric predictions of the drift-diffusion model, the subjects' reaction time (RT) was anti-correlated with the slope of the DV (Fig. 2f). Subjects made their decisions rapidly (Fig. S6), well within the 2 s limit.

Throughout the deliberation, the subjects maintained a fixation on a central target. Moreover, the subjects responded with a single movement on each trial (i.e., a SC or a BP choice). We validated this performance using electromyographic (EMG) recording of hand muscles and continual recordings of the eye gaze. Indeed, in all valid trials, deviations in horizontal eye gaze were only observed during SC choices (red in Fig. 2g), whereas increases in the hand EMG were only observed during BP choices (blue in Fig. 2h). This performance standard was maintained during the reversed task (Fig. S2).

**Broadband gamma activity reveals graded choices**
We collected data in subjects with electrodes implanted into brain regions implicated in decision formation, including parietal, frontal, premotor, and motor regions (Figs. S1 and S3e). We recorded neural

signals from these implanted electrodes throughout the decision process. We specifically evaluated broadband gamma (henceforth referred to as γ) activity, which has been shown to be tightly correlated with multi-unit spiking activity[47,48].

We first identified the effector-modulated regions, which showed modulation of γ activity around the time of choice compared with baseline during the SC and BP choices. We assessed the regions in which a modulation was observed during both choice kinds (SC&BP-modulated; Fig. 3a) as well as regions that showed modulation during a specific choice (SC-modulated, BP-modulated; Fig. 3b). The cortical areas with significant γ modulation are quantified in Fig. S3a, c.

The high temporal resolution of the intracranial recordings, together with their broad cortical coverage, allowed us to investigate the spatial-temporal dynamics of the forming decisions. We investigated these dynamics by averaging γ activity across all effector-modulated regions (Fig. S4c). We found that the average γ activity ramped up gradually (Fig. 3a, b, middle; Fig. S5a). Specifically, we found that the time course of γ activity correlated with the time course of the DV during each decision period (SC: average $R = 0.05$, $t(2319) = 9.4$, $p = 8.4 \times 10^{-21}$; BP: average $R = -0.06$, $t(2698) = -13.4$, $p = 8.1 \times 10^{-40}$; two-tailed t-tests). Furthermore, the distribution of reaction time showed a right-skewed trend (Fig. S6). While the gradually ramping γ activity is consistent with diffusion models of decision-making[14–16], the conclusion of the accumulation process is not. In particular, we found that γ activity was strongly graded by the

DV immediately prior to a choice (green bar in Fig. 3). This finding is at odds with the traditional formulation of the drift-diffusion model[16], which posits that neural activity at the time of choice reaches a bound instead of being graded.

We quantified this graded effect at the time of choice (green bar) using Spearman's correlation ($R$) between $\gamma$ activity and DV for data of each session. We found that the effector-modulated regions were significantly graded by the DV (Fig. 3a, b). The regions modulated by SC and BP choices had an average $R = 0.12$ and $R = -0.12$, respectively ($t(12) = 3.2$, $p = 7.3 \times 10^{-3}$; $t(12) = -4.1$, $p = 1.5 \times 10^{-3}$; two-tailed $t$-tests, Fig. 3a). The regions modulated by SC or BP choices showed similar effects (average $R = 0.12$, $t(12) = 3.2$, $p = 7.5 \times 10^{-3}$; average $R = -0.13$, $t(12) = -3.5$, $p = 4.7 \times 10^{-3}$; two-tailed t-tests, Fig. 3b). Furthermore, we found that BP-modulated regions were significantly graded by the DV at the time of SC choices, and the SC-modulated regions were significantly graded by the DV at the time of BP choices (Fig. S7a).

We evaluated the timing of this effect with respect to choice (Fig. 3c, d). For electrodes modulated by SC and BP, the correlation between $\gamma$ activity and DV at the time of choice was significant for the 300 and 240 ms prior to movement onset for BP and SC choices, respectively ($p < 0.01$, Fig. 3c). For electrodes modulated by SC or BP (Fig. 3d), the corresponding time was 300 and 40 ms prior to movement onset, respectively. Notably, the two plots show that the evidence accumulation process ceases following a choice. For both SC and BP choices, there is a marked decrease in the correlation values immediately following a choice. This result argues against potential post-processing that could take place during involved decisions.

### Specific test of the bounded hypothesis

We specifically tested the hypothesis that the neural signals that encode the forming decisions reach a fixed bound at the time of choice. The null hypothesis was set by a modeled DV that reaches a fixed bound at the time of each choice (Fig. 4a). We regressed the recorded $\gamma$ activity from all effector-modulated electrodes (same as Fig. 3a, b) on this modeled DV (Fig. S4d). If the null hypothesis was true, we would not expect a significant correlation between the regressed $\gamma$ and the raw DV at the time of choice. Yet, the graded effect was also prominent in this analysis (Fig. 4b). Specifically, we found that the regressed $\gamma$ activity ramped up as a function of time, but was strongly graded by the DV around the time of choice (green bar in Fig. 4b). We again quantified this graded effect by plotting the histogram of $R$ values over the individual sessions ($n = 13$, Fig. 4c). The regions modulated by SC and BP choices showed a significant session-average

of $R = 0.14$ and $R = -0.12$, respectively ($t(12) = 4.1$, $p = 1.5 \times 10^{-3}$; $t(12) = -4.7$, $p = 4.8 \times 10^{-4}$; two-tailed $t$-tests). The regions modulated by SC or BP choices showed a similar significant result (SC: average $R = 0.14$, $t(12) = 3.6$, $p = 3.9 \times 10^{-3}$; BP: average $R = -0.14$, $t(12) = -4.8$, $p = 4.3 \times 10^{-4}$; two-tailed t-tests). To control for the possibility of overfitting due to the high dimensionality of the linear model, we performed a randomization test in which we randomly shuffled the temporal relationship between the neural signals and the modeled DV. This control analysis shows no graded effect and therefore rules out an overfit (gray in Fig. 4c, Fig. S8). Thus, this analysis supports the alternative hypothesis that at the time of choice, the neural signals are graded by the DV.

### Graded signals across brain regions

We next investigated the specific brain regions that contributed to the decision-related effects. Specifically, we identified the brain regions that showed grading of $\gamma$ activity around the time of choice. For SC&BP-modulated regions (Fig. 5a; Fig. S3b), we found that BA40 (parietal cortex) has the most prominent contribution to the grading of the $\gamma$ activity. For SC/BP-modulated regions (Fig. 5b; Fig. S3d), we found BA40, BA8 (including frontal eye fields) for SC choice; and BA40, BA6 (premotor/supplementary motor areas) for BP choice have the most prominent contribution to the grading of the $\gamma$ activity.

The graded effect was apparent in $\gamma$ activity averaged across all SC&BP- (Fig. 5c) and SC/BP-graded (Fig. 5d) electrodes. Notably, this prominent effect was reliably observed in individual trials (see scatter plots). All $R$ values shown in the scatter plots were highly significant for SC and BP choices (SC&BP-graded regions in Fig. 5c: $R = 0.17$, $p = 4.3 \times 10^{-13}$ and $R = -0.15$, $p = 7.8 \times 10^{-12}$; SC/BP-graded regions in Fig. 5d: $R = 0.26$, $p = 3.3 \times 10^{-28}$ and $R = -0.19$, $p = 2.0 \times 10^{-21}$; SC&BP-graded parietal region in Fig. 5e: $R = 0.17$, $p = 2.9 \times 10^{-6}$ and $R = -0.16$, $p = 1.3 \times 10^{-6}$; two-tailed Spearman's correlation).

### The graded nature of the subjects' choices

The graded effect of $\gamma$ activity by DV at the time of choice suggests that subjects made their choices in a probabilistic manner, with varying levels of supporting evidence or certainty. To test the validity of this inference, we investigated how the subjects' choice probability was modulated by different levels of DV at the time of choice. As expected, we found that the level of evidence substantially modulated the choice probability (Fig. 6a) and that this graded modulation effect was significant for SC and BP choices ($F(2, 36) = 27.9$, $p = 4.8 \times 10^{-8}$; $F(2, 36) = 43.1$, $p = 2.8 \times 10^{-10}$; one-way ANOVA). Thus, the graded $\gamma$

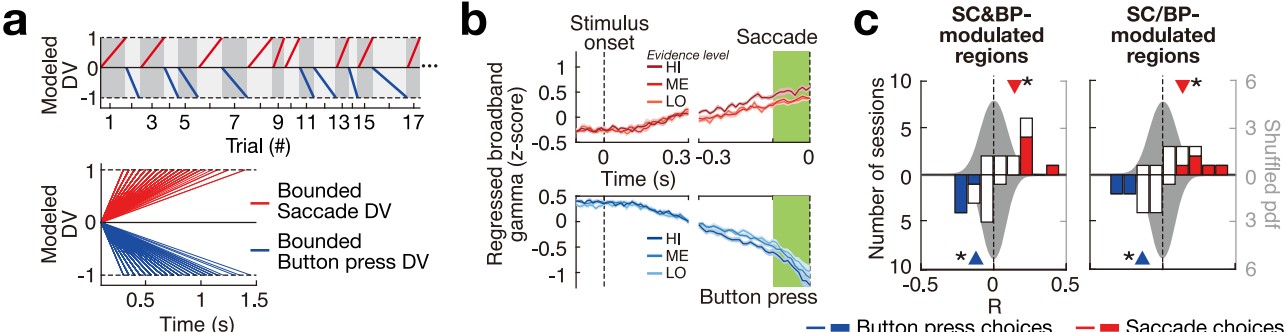

**Fig. 4 | Regression analysis rejects the null hypothesis of the signals reaching a fixed bound. a** Modeled DV with a fixed bound. The value of the modeled DV ramps up linearly until it reaches a fixed bound at the time of choice associated with each choice kind. **b** Regressed neural activity graded by DV at the time of choice. In this analysis, the broadband gamma ($\gamma$) activity of all effector-modulated electrodes (Fig. 3b) was regressed on the modeled DV as shown in (**a**). The regressed $\gamma$ is plotted on the ordinate as a function of time and averaged over the individual sessions (mean ± s.e.m., $n = 13$). The signals are aligned to stimulus onset (left

dashed line) and movement onset (right dashed line). The top (bottom) panel shows regressed $\gamma$ activity for trials that resulted in a SC (BP) choice. **c** Same format and analysis as in the histograms in Fig. 3. The shaded distribution results from a randomization test. Left panel: SC&BP-modulated regions. *$p = 1.5 \times 10^{-3}$ and $p = 4.8 \times 10^{-4}$ for SC and BP, respectively (two-tailed $t$-tests). Right panel: SC/BP-modulated regions. *$p = 3.9 \times 10^{-3}$ and $p = 4.3 \times 10^{-4}$ for SC and BP, respectively (two-tailed $t$-tests).

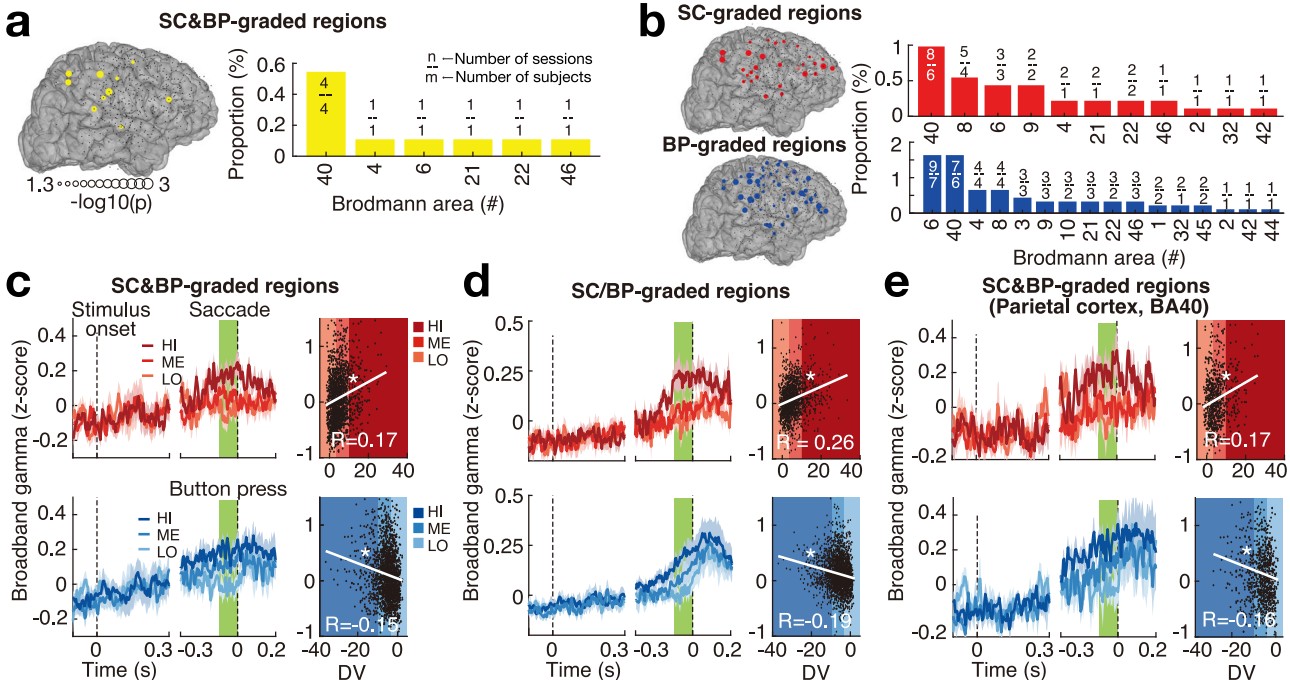

**Fig. 5 | Individual brain regions modulated by the DV at the time of choice and the decision dynamics. a** Left panel: The yellow symbols indicate electrodes with broadband gamma ($\gamma$) significantly ($p < 0.05$, one-tailed randomization tests) graded by the DV at the time of choice for both SC and BP choices (SC&BP-graded regions). Right panel: The bars show the proportion of electrodes within individual Brodmann areas (BA). The numbers above each bar indicate the number of sessions (numerator) and number of subjects (denominator). **b** Same format as in (**a**) but for electrodes with $\gamma$ significantly ($p < 0.05$, one-tailed randomization tests) modulated by DV at the time of choice during SC (red, SC-graded regions) and BP (blue, BP-graded regions) choices, respectively. **c** Left panel: Mean ± s.e.m. $\gamma$ activity within all SC&BP-graded electrodes ($n = 12$). Right panel: Single-trial effects. The mean $\gamma$

activity around the time of choice (green bars) on each trial is plotted against the corresponding DV value. *$p = 4.3 \times 10^{-13}$ and $p = 7.8 \times 10^{-12}$ for SC and BP, respectively (two-tailed Spearman's correlation). **d** Same format as in (**c**), but for signals specifically graded by the DV at the time of choice during the SC (top) and BP (bottom) choices within the SC-graded and BP-graded regions, respectively. The plots represent mean ± s.e.m. over the individual sessions ($n = 13$). *$p = 3.3 \times 10^{-28}$ and $p = 2.0 \times 10^{-21}$ for SC and BP, respectively (two-tailed Spearman's correlation). **e** Same format as in (**c**, **d**), but for BA40 (parietal cortex) electrodes graded by DV during both choices ($n = 6$). *$p = 2.9 \times 10^{-6}$ and $p = 1.3 \times 10^{-6}$ for SC and BP, respectively (two-tailed Spearman's correlation).

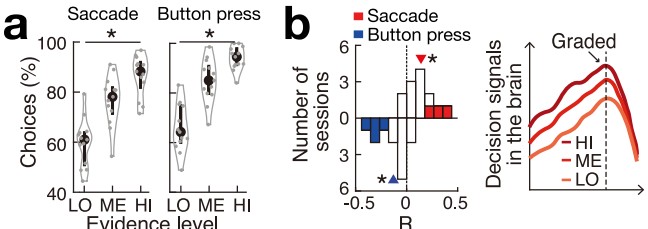

**Fig. 6 | Summary: The human brain can make analog choices. a** Choice probability ($n = 13$) for the three levels of decision evidence (low, medium, high) for saccade and button press (gray dots: each session; black dots: median across sessions). Violin plots show probability density, and vertical lines indicate the first and third quartiles. *$p = 4.8 \times 10^{-8}$ and $p = 2.8 \times 10^{-10}$ for saccade and button press, respectively (one-way ANOVA). **b** Graded cortical signals. Left panel: Fig. 3b-right. Right panel: Fig. 1b.

activity also represents the probability of producing the respective choice, indicating a continuum between the neural activity and behavior during decision-making.

If the decision-related $\gamma$ activity in the brain is related to the amount of evidence for or the probability of making a choice, efferent activity might also be detectable within the effector peripheral systems that execute each choice[49]. We found modest (Fig. S9) but significant support for an efferent effect. Specifically, the EMG activity of the hand muscles and the amplitude of the saccades made to the choice target

was significantly graded by the DV at the time of choice (BP: $t(12) = 4.0$, $p = 0.0017$; SC: $t(12) = 3.1$, $p = 0.010$; two-tailed t-tests).

Together, these data reveal that the human brain represented choices in an analog, graded manner (Fig. 6b) instead of an all-or-none, switch-like fashion.

## Discussion

In this study, we recorded LFP activity directly from the brain of human subjects making perceptual decisions. The recordings revealed that the human brain encodes developing perceptual decisions within $\gamma$ activity. Across response types (SC and BP), task contexts (congruent and reversed), and analysis methodologies (model-free and model-based), we found that the $\gamma$ activity remained graded at the time of choice, suggesting a graded instead of a definitive decision process.

Decision-makers live in dynamic environments with varying goals and behavioral demands. To make effective decisions, individuals must gather sensory evidence and use it to plan actions in a particular context. The context must consider relevant stimulus sensory information, relevant actions and their payoffs, and the mapping between the stimuli and the actions[33]. This contextual processing demands a flexible representation of the decision process. In this light, the dominant, bounded model (Fig. 1a) likely applies to a subset of decisions that are performed in stationary contexts, in which subjects make many similar decisions consecutively. In natural settings, decisions are almost exclusively made in new and dynamic contexts, which include variable decision policies that rest on factors such as urgency, reward expectation, speed-accuracy tradeoffs, stimulus-action

mapping, and decision confidence[33]. The graded model (Fig. 1b), in which neural activity does not need to reach a fixed level in the decision process, can accommodate these factors. Our task implements these dynamic contextual and task demands by varying the mapping between stimuli and actions (Fig. 2b). Indeed, subjects found the decisions to be challenging, with 55.2% of all trials classified as valid under our stringent acceptance criteria. Yet, the subjects were able to make robust evidence-guided decisions, demonstrated by a defined psychometric curve spanning the entire range of choice probabilities (Fig. 2e). In these more general, dynamic decision situations, we have found robust evidence for a graded nature of the decision process (Fig. 1b).

The intracranial electrodes covered a large portion of the cortex, which enabled us to inspect the regions contributing to the decision formation. The decision evidence was found to be encoded across multiple areas of the cortex, in particular parietal, frontal, and premotor regions (Fig. 5). This multi-regional representation is expected, given that the tasks required subjects to assess and integrate sensory evidence, compare it across two accumulating systems, and accordingly plan a movement. These aspects of the tasks have been shown to map onto the respective brain regions[50–58]. Moreover, the broad engagement of the nervous system during deliberation suggests that sensory, cognitive, and motor processes do not function in isolation but as a spatiotemporal continuum[42,43,50,59].

Many accounts of decision-making assume that decision evidence is tracked in a central cognitive module that is independent of the effector systems that execute the respective choices[60,61]. Contrary to this view, theories of embodied cognition have proposed that the decision-making process can be offloaded to cognitive faculties and neural circuits that implement the associated choices[62–66]. In our study, we found representations of decision evidence in many brain regions that encode choice, thus contributing to the theories of embodied cognition. Specifically, our data show that decisions can be formed within the same circuits that plan and execute the resulting choice. For example, the graded effect was most prominent over parietal (BA40) and premotor/supplementary motor areas (BA6) for BP choice; and over parietal (BA40) and frontal eye fields (BA8) for SC choice (Fig. 5). This distributed representation may preserve cognitive resources and accelerate the production of action[64–66].

The decision-related effect reported in our study cannot be explained by a potential confound of sensory activity throughout the task, for two reasons. First, we tested two distinct task contexts–congruent and reversed (Fig. 2b), thus swapping the mapping between stimulus and response (Fig. S2, Fig. S3f). Second, prior to performing the analysis, we excluded electrodes that were modulated by sound clicks (Fig. S1e), thus avoiding the potential confound of sound-induced neural activity.

The decision-related effect reported here cannot be explained by a potential confound of motor activity or response vigor, for five reasons. First, the effect was observed when the specific choice was fixed (sorting trials by the SC or BP choices), indicating that the graded effect was not an artifact of averaging potential purely motor signals (Figs. 3 and 4). Second, the graded effect was found in SC&BP-modulated regions and thus was not tied to a specific movement (Figs. 3 and 4). Third, the $\gamma$ activity ramped up gradually (Fig. 3, Fig. S5), reminiscent of similar findings in the parietal reach region and the lateral intraparietal region of non-human primates[67]. Fourth, the decision-related grading could be detected already 300 ms prior to a movement, and the effect was comparable across SC&BP-modulated and SC/BP-modulated regions (Fig. 3c, d). And fifth, the decision-related grading was strongest over the parietal cortex (Fig. 5), which represents higher-order cognitive variables rather than movements[15,62,68,69].

We found that the grading of the gamma activity was accompanied by the graded probability of producing the respective choice (Fig. 6a). This suggests that the grading may reflect a subject's certainty in their choice. This notion is supported by previous studies, which found that the signals representing accumulated evidence also encode a subject's certainty or confidence in their decisions[37,70].

There is emerging evidence that neural signals underlying perceptual decisions may not be limited by a bound. This evidence comprises electroencephalographic (EEG) recordings in humans[6–13] and single-neuron recordings in animals[1–5]. However, EEG recordings in humans predominantly capture low-frequency signals representing a summation of electric potentials across many brain regions. Consequently, EEG recordings are limited in their spatial resolution, and can only provide a generalized perspective on the underlying neural processes. On the other hand, single-neuron recordings in animals have been restricted to specific brain regions, hindering the ability to provide a conclusive answer to how the brain represents the entirety of the decision process. Here, we recorded LFPs directly from the human brain and evaluated localized high-frequency $\gamma$ activity, which is a surrogate of multi-unit discharge activity[47,48]. These neural signals provided direct evidence across multiple brain regions that the human brain can implement decisions in a graded manner. Even though multi-unit activity and LFPs are tightly correlated[47,48], and the standard model states that LFPs/EEG are the extracellular currents primarily reflecting summed postsynaptic potentials of pyramidal cells[71], the empirical literature linking EEG, LFPs, and microcircuit neural dynamics is under-explored[72–74]. Therefore, the findings of our study should be interpreted explicitly within localized, high-frequency neural discharges, and under the assumption that neuronal discharges constitute the primary code of decision-related neuronal signaling.

Our study has three limitations, which are common to decision tasks performed in laboratory settings. First, the study cannot distinguish exactly which factors (e.g., urgency, reward expectation, speed-accuracy tradeoffs, etc.) the DV-related $\gamma$ activity represents. The graded $\gamma$ activity could represent the DV itself, the confidence in a choice, or any other correlated variables. As such, the graded effects reported here should be interpreted as decision-related. Nonetheless, regardless of which exact decision-related variable is represented, our study highlights its graded nature. Second, although it has been argued that sensory-motor decisions studied in the laboratory setting are likely based on the same deliberative processes as decisions encountered in real life[15], this assertion remains to be demonstrated by recording neural activity in naturalistic decision-making scenarios[33]. Our study, which engaged human subjects with varying mapping between evidence and effectors, takes a step in this direction while still tightly controlling for the temporal aspect of the evidence accumulation. Third, it is important to stress that only a subset of the recorded regions showed modulation by the DV at the time of choice. It is, therefore, possible that there are other regions involved in evidence accumulation towards a bound besides those inspected here. Implanting electrodes into many more brain regions (including deep brain regions) would help to address this matter.

In summary, we report neural evidence for a graded nature of the decision process. Intracranial recordings suggest that the human brain possesses the capacity for processing evidence and making choices in a flexible, analog manner. Such analog decision-related representations do not require an all-or-none conversion, which may constitute a substrate for the flexible choice behavior common to human decision-makers.

## Methods

### Subjects
Local field potential (LFP) neural activities were recorded using intracranial electrodes implanted in 8 humans (5 males, 3 females, 6 right-handed, 2 left-handed, aged 15–57 (mean ± s.d. age of 39 ± 15), recorded from July/2012 to February/2020; Fig. S1). The subjects underwent

surgery for temporary placement of subdural grid electrodes (7 subjects) or intracerebral stereotactical electrodes (1 subject) to localize their epileptogenic focus. All subjects had normal cognitive capacity (mean ± s.d. IQ of 94.5 ± 17.2), normal hearing, normal or corrected-to-normal vision, and were able to perform the highly controlled task. The grid electrodes consisted of platinum-iridium contacts, which were 4 mm in diameter (2.3 mm exposed) with an inter-electrode distance of 6 or 10 mm. The stereotactical electrodes consisted of platinum/iridium contacts 0.8 mm in diameter and spaced 3.5 mm apart (contact length 2 mm, insulation length 1.5 mm). We limited our analysis to electrode contacts distant from epileptic foci, and used contacts distant from areas of interest for reference and ground. The study was approved by the Institutional Review Board of Albany Medical College. All subjects gave informed consent to participate in the study. For subjects below the age of 18 years, informed consent was obtained from their legal guardians.

## Data Collection

The subjects were positioned in front of a flat-screen monitor that presented the visual stimuli (17" diagonal size, 60 cm distance). The auditory stimuli were delivered using headphones (MDR-V600, Sony). The stimuli consisted of a train of brief (0.2 ms) click sounds drawn from a homogeneous Poisson distribution. The stereo stimulus was composed such that the sum of clicks presented to the left ear ($C_l$) plus the sum of clicks presented to the right ear ($C_r$) summed to a fixed number $C_l + C_r = 50$ in 2 s. Each train lasted for up to 2 s. Consecutive clicks were spaced by at least 5 ms, and the initial click in each trial occurred in both ears simultaneously[46]. For each train, the subjects determined whether they perceived more clicks in the ear contralateral or ipsilateral to the intracranial recording hemisphere. They were free to indicate their choice during any time of the auditory presentation via the joystick or saccade. The auditory stimulus ceased upon a choice. The hand contralateral to the recording hemisphere rested on a pillow placed on a fixed table while holding a joystick (Logitech Attack 3). Subjects were instructed to simultaneously press the front and top buttons using their index finger and thumb, respectively. We used the two-finger response to potentially engage a wider network of the movement planning circuitry compared to if we had only used the response of a single digit. Additionally, we recorded surface electromyographic (EMG) activity from anterior forearm muscles to track the muscle activity during each decision. In total, we placed five surface electrodes (pre-gelled disposable Ag/AgCl EMG electrodes) on the forearm muscles. Four electrodes were arranged as a 2 × 2 grid with an inter-electrode distance of 2 cm horizontally and vertically. They were placed on the flexor digitorum superficialis muscle. One electrode was placed on the first dorsal interosseous muscle. An additional EMG electrode was placed on an electrically neutral tissue as the ground for EMG recording. The eye gaze position of each eye was measured 60 times/second using an eye tracker (Tobii T60, Tobii Technology) integrated into the flat-screen monitor. Neural signals and EMG signals were simultaneously amplified and sampled at 1200 Hz in a manner that prevents aliasing (g.USBamp/g.HIamp biosignal acquisition devices, g.tec). Synchronized acquisition of neural signals, EMG signals, eye gaze, joystick responses, and task control (Fig. 2a) was accomplished with BCI2000[75].

## Task

Each trial started with the visual presentation of a red fixation cross, 2 visual degrees in size (Fig. 2b). Subjects had to maintain fixation within 2 visual degrees. After acquiring fixation, two icons appeared 15 visual degrees to the left and right to the fixation cross. The icons and auditory stimuli were presented on the sides contralateral and ipsilateral to the recording hemisphere. The icon at the contralateral side was a sketch of a joystick. The icon at the ipsilateral side was a sketch of an eye. At the same time, subjects listened to a binaurally presented

auditory stimulus with a maximum duration of 2 s. Subjects had to determine whether they heard more clicks in the contralateral or ipsilateral ear. Subjects were free to indicate their choice during any time of the auditory presentation via the joystick or saccade. The auditory stimulus ceased upon a choice. In the congruent task context (Fig. 2b, top), if subjects heard more clicks in the contralateral ear, they simultaneously pressed the front and the top buttons of the joystick. In contrast, if subjects heard more clicks in the ipsilateral ear, they directed a saccade to the eye icon on the ipsilateral side of the monitor. This contingency was flipped in the reversed task context (Fig. 2b, bottom). Hand movement onset was taken as the time at which the first button was pressed. The eye movement onset was taken as the time at which the eye gaze started moving away from the fixation target with a certain velocity criterion (2% of the maximum velocity). We corrected the eye movement onset by a measured 33 ms latency of the Tobii T60 eye tracker. If subjects broke fixation for more than 150 ms, pressed any button before the auditory stimulus onset, responded with both button press and eye movements, or failed to indicate a response within 2 s following the stimulus onset, the trial was aborted and excluded from analyses. The type of failed responses was indicated to the subjects in red and large-font text (fixation break: TOO EARLY; no response: TOO LATE; response with both movements: MOVED BOTH). A successful choice was communicated to the subject by shrinking the icon corresponding to the choice (the eye icon or the joystick icon) from 2 visual degrees in size to 1 visual degree in size. After subjects re-acquired fixation and released all buttons, they were given visual feedback for 0.65 s indicating whether they were correct. A correct response was indicated by a green text (+20c), while an incorrect response was indicated by a red text (−20c). The offset of feedback was followed by a 0.1 s inter-trial interval. We collected n = 7 sessions in the congruent task and n = 6 sessions in the reversed task. Subjects performed only one task type (congruent or reversed) in each session. Specifically, 7 subjects performed the congruent task, 6 subjects performed the reversed task, and 5 subjects performed both tasks (Fig. S1c). Overall, an average of 212 (84–490) and 245 (90–434) trials per session were analyzed during saccade (SC) and button press (BP) choices, respectively (mean (min–max), n = 13). Across the sessions, valid trials constituted 55.2 ± 18.1% (mean ± s.d, n = 13) of all trials. This modest proportion reflects the highly controlled nature of the task. The incorrect response trials were further excluded to eliminate the potential confound related to error response (Fig. S1c). Importantly, to avoid the potential mixing of decision difficulty and response type, we have separated SC and BP trials throughout the manuscript. The analysis of the congruent and reversed data was also performed separately.

## Decision variable

To capture the choice behavior of subjects in this task, we devised a decision variable (DV) according to signal detection theory[15]. In particular, a simple DV constructed from discrete, independent pieces of evidence (click sounds) can be evaluated using the logarithm of the likelihood ratio of either choice (Fig. 2c):

$$DV(t) = \log LR(t) = \sum_{i=1}^{i=t} \log \frac{P(e_i|SC)}{P(e_i|BP)}, \tag{1}$$

where the sum runs from the first click ($i = 1$) up to the last click ($i = t$) occurring prior to or at time $t$; $e_i$ is the $i$-th click (right- or left-click); $P(e_i|\text{choice})$ is the probability of $e_i$ in a given choice. For example, $P(e_i(\text{left-click})|SC)$ is the probability that a click is a left-click given a SC choice and analogously for the 3 other combinations of the arguments of $P$ (i.e., $P(e_i(\text{left-click})|BP)$, $P(e_i(\text{right-click})|SC)$, $P(e_i(\text{right-click})|BP)$). These probabilities were computed from the frequencies of the summed left (or right) clicks over all trials of a given choice[70]. We define the DV at the time of choice as the DV value 100 ms preceding

movement onset (Fig. S4b), and grouped the data for a given correct choice into terciles of growing evidence for that choice, i.e., LO (low), ME (medium), and HI (high). Therefore, each evidence level has the same number of trials for a given choice kind in each session. We further calculated the DV slope by using a linear fitting of DV within the range of 200 ms after stimulus onset to the time of choice.

## Data preprocessing

Prior to performing analyses, we visually inspected the signals recorded from all electrodes and excluded those that exhibited epileptic and artifactual activity (228/799). Together, 571 electrodes in eight subjects were included in the analyses (Fig. S1). Electrodes involved in the congruent task session analysis were identical to those in the reversed session. We established the anatomical locations of the implanted electrodes by co-registering the post-operative computer tomography image (CT) with a pre-operative magnet resonance imaging (MRI) T1 image. We used FreeSurfer (version 7.2.0)[76] to generate a 3D model of the cortical surface. Next, we projected the electrodes of all subdural grid subjects onto the surface of the cortex, accounting for eventual brain shifts caused by the craniotomy. To determine the Brodmann areas (BA) in which the electrodes are located, we first transformed the electrode locations into Talairach space, using a standard AC-PC alignment approach[77]. Next, we labeled each electrode based on annotations provided by Talairach Demon[78]. We analyzed the EMG signals using the bipolar configuration, notch filtering (60 and 120 Hz) the signals, band-pass filtering them between 20 and 170 Hz, and computed the envelope by taking the absolute value of the Hilbert transform applied to the signals. All signal analyses in our study are performed in MATLAB (2017b, Mathworks, Inc., Natick, MA). We used Hamming-windowed sinc FIR filters for all the filtering processes (`pop_eegfiltnew()`, EEGLAB, version 2022.0), which performs forward-backward filtering to avoid time shifts by using the `filtfilt()` function in Matlab.

## Extracting the broadband gamma ($\gamma$) signals

The intracranial neural signals recorded from the brain were high-pass filtered at 0.5 Hz. Spatially distributed noise common to all electrodes was removed using a common average reference filter. We applied notch filters to remove line noise at 60 and 120 Hz. We analyzed brain signals within the canonical broadband gamma ($\gamma$) range (Fig. S4a). Specifically, we filtered the neural signals from each electrode at 70–170 Hz, and computed the envelope by taking the absolute value of the Hilbert transform of the resulting signal. We further z-scored the $\gamma$ signals of each electrode.

## Extracting the auditory-related electrodes

To determine auditory selective locations, the subjects performed an additional passive listening task. Specifically, subjects listened to short stories presented with computer speakers, while neural activity was recorded. In each trial, BCI2000 cued the subject to the task by presenting the words "listen carefully" or "stop and relax". Each short story lasted for 17–36 s, and was followed by a resting period of 15 s. Overall, an average of 32 (15–53) trials per subject were analyzed. We performed the same procedure as above to obtain the $\gamma$ signals.

We assumed that the auditory-related electrodes should show a significant $\gamma$ increase for both the decision task (Fig. 2b) and the passive listening task. For the decision task, we computed the mean $\gamma$ during the baseline period (250 ms window preceding stimulus onset) and stimulus period (50 ms to 300 ms following stimulus onset) for each trial. For the passive listening task, we computed the bin mean (1-second bin) of the $\gamma$ during the baseline period (14 s preceding stimulus onset) and stimulus period (14 s following stimulus onset) for each trial. To determine whether an electrode had a significant auditory response, we correlated the obtained mean values with a vector of condition labels (baseline period = −1,

stimulus period = 1), and performed a biserial rank correlation, which provided a Spearman's $R$ value for each electrode. We tested the significance of the obtained $R$ values against a shuffled null distribution of $R$ using a randomization test. Specifically, the condition labels vector was randomly reordered (without replacement) and a newly computed $R$ value. This process was repeated 1000 times. In all cases, we tested for the normality of the null distribution using the Kolmogorov–Smirnov test. We then determined the probability that a given tested $R$ originated from the respective null distribution. This probability constituted the resulting $p$ values. We corrected the resulting $p$ values for the number of electrodes in each subject using a false discovery rate (FDR). Electrodes with significant ($p < 0.05$) $\gamma$ increase for both the decision task and passive listening task were defined as auditory-related electrodes.

To further eliminate the possibility of a purely auditory response, we excluded 79/571 auditory-related electrodes from eight subjects (Fig. S1e). This has provided 492 electrodes in eight subjects for analysis.

## Identifying the effector-modulated regions

To identify the effector-modulated electrodes, for SC (BP) choice trials of each electrode, we computed the mean of the $\gamma$ activity during the baseline period (50 ms to 300 ms following stimulus onset) and effector-related period (200 ms preceding to 50 ms following movement onset). Next, we correlated the obtained mean values with a vector of condition labels (baseline period = −1, effector-related period = 1), which provided a Spearman's $R$ value. We performed the same randomization test as above and obtained a $p$ value (FDR corrected for the number of electrodes in each subject). Electrodes showing ($p < 0.05$) significant difference (increase or decrease) in $\gamma$ activity at response relative to baseline were identified as effector-modulated electrodes. Specifically, among the effector-modulated regions, we found that ($72 \pm 24$)% and ($73 \pm 18$)% of electrodes (mean ± s.d., $n = 13$) showed significantly increased $\gamma$ activity for SC and BP choices, respectively. The remaining effector-modulated electrodes showed significantly decreased $\gamma$ activity. We included all effector-modulated electrodes in the analysis to prevent biased selection. Electrodes showing significant $\gamma$ modulation for both SC and BP choices were defined as SC&BP-modulated electrodes (Fig. 3a). Electrodes showing significant $\gamma$ modulation for SC choices were defined as SC-modulated electrodes, and significant $\gamma$ modulation for BP choices as BP-modulated electrodes (Fig. 3b). Electrodes showing significant $\gamma$ modulation in one effector type but not in another were defined as effector-selective electrodes (Fig. S7b). The effector-related $\gamma$ modulation responses were independent of the DV.

## Model-free analysis

The key test in our study was the evaluation of neural signals graded by DV at the time of choice (referred to as DV-graded henceforth). To evaluate the graded effect across the SC&BP-modulated regions, we averaged the $\gamma$ activity of SC&BP-modulated electrodes, and computed the correlation between the averaged $\gamma$ with the DV at the time of choice (Fig. S4c, Fig. 3). Specifically, we calculated the mean value of the averaged $\gamma$ activity over a 100 ms window preceding movement onset for each trial. Next, we correlated the obtained SC (BP) mean values with the SC (BP) DV at the time of choice (Fig. S4b), resulting in a Spearman's $R$ value. We assessed the significance of this $R$ value using a randomization test. In this test, the obtained mean values were randomly reordered (without replacement) and a new $R$ value was computed, and this process was repeated 1000 times. We assessed the significance of correlation using the same procedure as above (results showed in Fig. 3a, b, right panel bar). To evaluate the graded effect across the SC-modulated or BP-modulated regions, we averaged the SC and BP $\gamma$ modulated electrodes, respectively, and performed the same correlation analysis.

## Model-based analysis

Based on the drift-diffusion model in which the cognitive processes terminated towards a choice when the decision-related neural activity hit a fixed bound, we made a null bounded hypothesis that the DV would reach a bound for each decision Fig. 4a. We tested the DV-graded effect using a linear regression model based on this null hypothesis (Fig. S4d). Specifically, we generated a modeled SC (BP) DV with the value linearly ramped from 0 to 1 (−1), spanning between 100 ms following the stimulus onset to 100 ms preceding movement onset. Additionally, the modeled SC (BP) DV value during the 100 ms window preceding the movement onset was 1 (−1), with the remaining part being 0. We down-sampled (`pop_resample()` in EEGLAB) the $\gamma$ activity to 100 Hz and regressed (`regress()` in Matlab) the $\gamma$ from effector-modulated electrodes onto the modeled SC (BP) DV. The null hypothesis was considered true if there was no significant correlation between the regressed $\gamma$ activity (i.e., the predicted values from the regression model) and the raw DV at the time of choice. Rejection of the null hypothesis could confirm the DV-graded effect.

To evaluate the graded effect across the SC&BP-modulated regions, we regressed the $\gamma$ activity from SC&BP-modulated electrodes onto the modeled SC (BP) DV. The time window used for the regression spanned the period from 200 ms preceding the stimulus onset to the onset of a movement. This regression provided a set of weights, which enabled us to predict the modeled SC (BP) DV inferred from the $\gamma$ at each moment in time. Next, we calculated the mean value of the regressed $\gamma$ over a 100 ms window preceding movement onset for each trial, and correlated the obtained SC (BP) mean values with the raw SC (BP) DV at the time of choice (Fig. S4b), resulting in a Spearman's $R$ value. We assessed the significance of this $R$ value using the same randomization test in the model-free analysis. To control for the possibility of overfitting due to the high dimensionality of the linear model, we performed a randomization test in which we randomly shuffled the temporal relationship between the neural signals and the modeled DV. Specifically, the $\gamma$ from all SC&BP-modulated electrodes were simultaneously circular-shifted in time by a randomly selected amount, and a new $R$ value was computed using the same regress process. We repeated this process 1000 times, obtaining a null distribution of $R$. We assessed the significance of correlation using the same procedure as above (results shown as shaded distribution in Fig. 4c). Notably, the circular-shift is a more stringent test than random shuffling as it leaves the temporal structure and thus the autocorrelation of the $\gamma$ intact (it only abolishes the temporal relationship between the $\gamma$ and the DV).

To evaluate the graded effect across the SC/BP-modulated regions, we regressed the $\gamma$ from SC-modulated and BP-modulated electrodes onto the modeled SC DV and BP DV, respectively.

## Extracting electrodes graded by DV at the time of choice

We found a graded effect for both SC and BP choices in SC&BP-modulated and SC/BP-modulated regions (Figs. 3 and 4). Specifically, the $\gamma$ signals were positively modulated by the DV at the time of choice. To further evaluate the individual brain areas that contributed to this DV-graded modulation effect, we calculated the mean value of the $\gamma$ over a 100 ms window preceding movement onset for each trial and each effector-modulated electrode. Next, we correlated the obtained SC (BP) mean values with the SC (BP) DV at the time of choice (Fig. S4b) and performed the same randomization test as the Model-free analysis. Electrodes with the $\gamma$ significant ($p < 0.05$) positively graded by SC (BP) DV at the time of choice were defined as SC (BP)-graded electrodes (Fig. 5b, Fig. S3d). Electrodes positively graded by both SC and BP DV were defined as SC&BP-graded electrodes (Fig. 5a, Fig. S3b).

## Reporting summary

Further information on research design is available in the Nature Portfolio Reporting Summary linked to this article.

## Data availability

Full datasets from our clinical recordings will be provided to interested researchers upon request to the corresponding authors and institutional approval of a data-sharing agreement. Source data are provided with this paper.

## Code availability

The custom analysis code used to generate our results will be available upon request to the corresponding authors.

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

## Acknowledgements

This work was supported by the National Institutes of Health (NIH) grants R01-MH120194 (to J.T.W.), R01-EB026439 (to P.B.), U24-NS109103 (to P.B.), U01-NS108916 (to P.B.), U01-NS128612 (to P.B.), P41-EB018783 (to P.B.), R00-NS100986 (to J.K.), RF1-NS128569 (to J.K.). T.X. was supported by the Brain & Behavior Research Foundation (NARSAD Young Investigator) and the McDonnell Center for Systems Neuroscience. P.B. was also supported by the McDonnell Center for Systems Neuroscience and the Fondazione Neurone. J.T.W. was also supported by the American Epilepsy Society fellowship award (Research and Training Fellowships for clinicians).

## Author contributions

J.K., T.X., J.T.W., and P.B. conceived of the study. T.X. and J.K. conceptualized the data analytical approach. M.A.A., A.L.R., and P.B. contributed to data collection. M.A. performed electrode localization procedures. H.C. contributed to the model-based analysis. T.X. and J.K. performed data analyses. T.X. and J.K. wrote the first draft of the manuscript, and all authors contributed to the writing and revision of the manuscript. J.K. and P.B. supervised all aspects of the study.

## Competing interests

The authors declare no conflict of interest.
