## [Peer Review File · Nature Communications]

Graded decisions in the human brainREVIEWER COMMENTS

Reviewer #1 (Remarks to the Author):

A large body of previous neurophysiological investigations in humans and non-humans have highlighted decision signals that exhibit a threshold-crossing relationships with choice behaviour, reaching a stereotyped amplitude immediately prior to a choice report. The present paper sought to determine whether the brain carries signatures of the evolving decision that are graded, right up to the point of commitment, consistent with a representation of confidence. Intracranial EEG data were collected from 8 pre-surgical patients performing a perceptual discrimination task (poisson clicks) in which saccades and hand button presses were used to report choices and stimulus-response mappings were varied across sessions. Data analyses centered on gamma band activity. The authors first identified sensors showing significant increases in activity at response relative to baseline for both response modalities (termed 'effector-independent') and those that only showed activity increases for one modality (termed 'effector-specific') and then examined the degree to which pre-response activity at these sensors scaled with the estimated DV extracted from the physical stimulus sequence. The results highlight a distributed set of sensors exhibiting a graded relationship with the DV covering primarily parietal, premotor and prefrontal areas.

I found this paper a very interesting read and the data are clearly of very high value. Although there are only 8 patients included, there was excellent sensor coverage of the cortex which is a key strength of the paper. I was also impressed with the authors efforts to exclude purely sensory-evoked signals by first eliminating any sensors that were activated in response to the clicks during passive listening. However, I do have a number of significant concerns about key aspects of the data analysis methods as well as with the framing of the paper with respect to the prior literature.

Major Comments

1. I found the rationale for the paper as framed in the Abstract and Introduction very confusing. It feels to me like a straw man argument to suggest that the field is in the midst of a debate between those who believe that the brain makes decisions in a purely all or none fashion and those who believe the brain carries graded representations of choice confidence. I have not come across anyone who would hold to the former view. It is true that the standard drift diffusion model assumes a fixed bound at response but numerous alternative models exist that allow for time-varying bounds that are not acknowledged by the authors. For example, there is now strong evidence from neurophysiological investigations and modelling studies supporting the role of collapsing bounds or urgency in decision making (e.g. Hanks et al 2014, eLife; Murphy et al 2016, Nat Comms) whereby the brain progressively lowers the cumulative evidence required to trigger commitment as time passes. Further, relationships between pre-response decision signal amplitudes and cumulative evidence have been reported in non-invasive human brain recordings

(e.g. Kelly et al 2021, Nat Hum Beh; Steinemann et al 2018, Nat Comms) and direct recording studies in monkeys have demonstrated a graded relationship between the spike rates of decision variable encoding neurons and choice confidence (e.g. Kiani et al 2009). So with all this in mind it is hard to discern exactly what theory or hypothesis the authors are seeking to test here. It seems to me like the authors would be better off focussing on the goal of localising brain regions that exhibit certain decision-related characteristics given that the human brain's decision making architecture remains quite mysterious and has proven difficult to definitively map through non-invasive means.

2. The introduction also makes much of the fact that signals exhibiting threshold-crossing relationships to behaviour have tended to be observed in experiments with highly-trained participants. The authors note then that a goal of the study was to examine decision signals in a context where participants have not had much exposure to the task. This aspect of the paper is treated in quite a vague way. First, it is not clear what the authors consider 'highly trained'. Their participants underwent hundreds of trials worth of testing so that would appear to be pretty comparable to most human investigations of decision making. Second, the authors appear to be implying then that the reason they do not see threshold-crossing effects in their data is because the participants had not been extensively trained. This claim is just not supported by the data for two reasons. First, because the authors have not conducted a systematic manipulation of task exposure and, second, because the authors have not conducted analyses that could rule out the possibility that threshold-crossing signals may be present in their data. As far as I can see, no analyses are provided to identify sensors that exhibit a threshold-crossing effect (e.g. minimum amplitude variance in the pre-response measurement window) and the focus on searching for sensors that correlate with the DV would exclude such signals by definition. As a more minor point, if the authors were to have strong evidence for the absence of threshold-crossing signals in their recordings, then they should provide some Discussion around the fact that this rules out what has been assumed to be the key stopping mechanism for the brain's decision process. In other words, if the signal identified in this study have a graded relationship with the DV then what determines the point at which commitment is reached and evidence accumulation terminated?

3. The criteria for classifying sensors as 'effector-independent' and 'effector-selective' appears problematic. It seems highly likely that motor preparation areas that are selective for hand movements might still exhibit some activity during decisions that were ultimately reported via saccades and vice versa - in fact it would be very surprising if this were not the case. According to the authors' current approach such signals would end up being classed as 'effector-independent'. The authors do show using eye tracking and EMG that there is no apparent systematic activation of the unused effector but this does not mean that we can assume that there is no preparation of these responses at the level of the cortex and many models and previous neurophysiological observations would predict that there should be some accumulation of evidence for the unchosen alternative. A plot showing gamma waveforms for saccade-selective sensors on trials that were reported with button pushes (and vice versa) would be really important to include. Regardless, it doesn't seem appropriate to class a sensor as 'effector-independent' when it's possible that a sensor could show significant activation for both response types and yet show significantly greater activation for one response-type than the other.

4. A total of 13 sessions were recorded in the two tasks and the key statistical tests are performed across sessions rather than participants. Given that only 55% of trials were valid there is a need to establish whether or not certain trial types and participants were more prominent in some of the cells of the analysis. For example, can the authors exclude the possibility that the signal scaling with DV is not down

to different mixtures of saccade vs button responses across difficulty? This is a particular concern because there appear to be systematic differences in the slope of the relationship between RT and DV for saccade vs manual responses. Similarly can the authors rule out confounds due to imbalances in the number of congruent vs incongruent trials across difficulty levels or variations in the presence/absence of certain sensors in a given session?

Minor Comments:

In the introduction the authors state “This debate has been difficult to resolve at the general level because there has been no study that has i) recorded neural activity at high spatiotemporal coverage directly from the brain”. The authors would need to clarify why direct recordings are necessary to resolve the ‘debate’ given that graded signals have already been identified in non-invasive recording studies.

The authors should provide more discussion of the distinct functional roles that the identified effector-independent and effector-selective areas might be playing in the decision process.

“The graded modulation during both choice kinds (effector-independent, Fig. 5a-top; Fig. S5) was strongest in the parietal cortex (BA 40), while the graded modulation during each choice (effector-specific, Fig. 5b-top; Fig. S5) was most prominent over parietal (BA 40) and premotor/supplementary motor areas” This is not really apparent from the plot it looks like there is a lot of overlap in the distribution of sensors for both signal types

Discussion – “Within the traditional drift-diffusion model , a subject could meaningfully set a putative bound (Fig. 1a) only after having learned the statistics of the given decision task. This would require an extensive exposure to the particular task.” I don’t follow the authors’ point here. There is nothing to stop a participant setting a bound on their response right from the very first trial, it’s just that they would require training in order to identify the optimal bound.

Reviewer #2 (Remarks to the Author):

This is an interesting and somewhat difficult paper that challenges the commonly-held belief that choices at the neural level can be modeled as an evidence accumulation process with a fixed boundary that represents an internal criterion.

Using a paradigm previously developed by Kubanek et al (NeuroImage 2013) and high-resolution ECoG recordings, the authors show compelling evidence that the evidence accumulates beyond a decision threshold and that the peak of evidence accumulation is directly correlated to the evidence itself (e.g., clearer signals reach higher boundaries). This implies that not only the relative evidence for different options but the decisions themselves are graded (“analog”) rather than binary and thresholded.

If confirmed, this finding is revolutionary. The idea of evidence accumulation to a threshold, as codified in the vast family of accumulator and drift-diffusion models, has become one of the cornerstones of the cognitive neurosciences, confirmed in multiple experiments in humans (e.g., the work of Forstmann) and nonhuman primates (e.g., Shadlen).

POSITIVES

The manuscript is well written. The data is of high quality, coming from 13 sessions with 8 patients with ECoG recordings. Multiple analyses are provided to show and bolster the points made. The distinction between effector-dependent and effector-independent regions is a particularly solid point, together with the correlation analysis between evidence and gamma activity.

In the discussion, the authors address some of the most obvious counterpoints, including the possibility that gamma activity tracks sensory representations rather than choices.

The analysis of Figure 3, in which linear ballistic evidence accumulation is regressed out, is particularly elegant and persuasive.

NEGATIVES

The accumulation-to-threshold framework is not online intuitive and appealing but has robust empirical support in the literature finding separate neural correlates of accumulation rates and thresholds. So, it would be interesting to know what the authors could offer as an explanation for why the graded nature of evidence has not been found before. Some of the original findings, in fact, clearly show evidence accumulation reaching identical thresholds for different trials (most famously, Roitman & Shadlen 2002). These older findings were established with single-cell, spike-train recordings but, as the authors point out, cortical gamma activity is supposed to track these single-cell spikes at an aggregate level. It is hard to compare the results exactly. In Roitman and Shadlen's paper, for example (as well as in Shadlen et al., 2016 Science), firing rates are visualized in response-lock fashion but also divided into classes of

response times (300ms, 400ms, and so on), and the accumulation curves, as expected by the theory, are steeper for shorter RTs, where evidence accumulation would have been faster. Because no such analysis is presented, it is hard to draw conclusions. (In fact, I saw no analysis where RTs are provided for the high, medium, and low evidence signals)

Similarly, the diffusion models with a single threshold would require that the accumulation curves for different conditions (high, medium, and low evidence) start at identical points at the stimulus onset, but then to ramp up at different slopes. This does not seem the case in Fig. 3; however, to judge this, the stimulus-locked and response-locked plots of gamma activity in Figure 3 would need to extend beyond 200ms and or -200ms.

A possible explanation that I do not see ruled out is the following. The paradigm used in this study, although established in the field and use for over a decade, is somewhat peculiar: The two left/right response options are mapped to different effectors (saccade vs. fingers). Of course, in any 2AFC task, there must be two distinct possible behaviors to indicate the preferred response. However, the most typical paradigms use extremely easy, consistent, and compatible mappings (left/right choice → left/right saccade). Even with these compatible mappings, and especially with animal models, subjects are extensively trained before the experiment. Both compatibility and training reduce the time necessary to translate the internal decision to the corresponding action. If indeed such a decision takes a significant amount of time (let's say, > 100ms), evidence would continue to accumulate in that period even if a decision has been made. In other words: If the mapping requires a significant second processing stage after the decision threshold has been reached, it would seem like the thresholds are different when they are not. A good way to rule this out would be to visualize a larger window of the gamma-activity response-locked plots in Fig. 3, and visualize the trials by RT bins (200-300ms, 300-400ms, 400-500ms, and so on; like in Shadle et al 2016 Science). If there a significant second stage before the motor/saccade response, the traces should appear to cross over in that window.

A second possible explanation is that the graded accumulation is a side effect of the nature of the task. Drift diffusion models assume that evidence might fluctuate randomly between two choices before two decisions are made. Virtually every paradigm I can think of (e.g., random dot motion, whiteness contrast...) uses stimuli where the stimulus does not really change over time, the drift is assumed to be due to noisy sampling, and participants are free to respond whenever they have sampled sufficient evidence. However, as far as I understand this specific paradigm, it is possible that clicks accumulate unevenly for the left and right ear (and Figure 2C seems to support that) and the clicks are provided for a fixed 2s period. In this case, participants might be following a very different strategy, waiting for the entire 2-second period before making a decision. In this case, one would naturally expect that different stimuli would be associated with different amounts of evidence, because, in fact, participants would not be using a threshold at all – they would simply wait until the end and pick the location with the highest momentary evidence. To rule out this possibility, it would be important to have plots that showcase the distribution of response times as well as accuracies and gamma activities because response times would give an indication of the strategy used. If the majority of response times are concentrated on or after the

2-sec mark, then it would be evidence that participants are not using an internal threshold as a decision, but are simply waiting for the clicks to end.

Finally, while the authors provide evidence against the single-threshold model, it is not clear what they would suggest as a replacement. At times, the text seems to imply that they think that the probability of a response simply grows with the corresponding gamma activity, without a definite threshold—sort of a threshold-less drift process? Is this the case that the authors are making? If so, I think it would be important to make it clear how they think that a decision is made. Of course, a full model would be outside the scope of the paper, but it would be interesting to read how the authors think about a viable alternative for how decisions happen.

MINOR

I think I understand the correlation analysis, but I am puzzled by some statistics—in particular, the passage from “ $R = 0.12$ and $R = -0.12$ during the saccade and button press choices” to

“ $t(12) = 3.2$, $p = 7.3 \times 10^{-3}$ ”; I understand that R values are means across many trials, but how is such a large t -value derived from their distribution?

Reviewer #3 (Remarks to the Author):

This study examined the temporal dynamics of electrical activity from implanted electrodes in people they performed an auditory perceptual choice task. To test whether activity prior the a decision is based on graded or threshold (boundary) processes, the authors evaluated broadband gamma activity averaged across electrodes in parietal, frontal, premotor, and motor regions. The task was to identify the laterality of auditory clicks presented to either ear. Click density was randomly chosen from a distribution but later assigned to high, medium, and low to indicated strength of evidence. Subjects responded using button presses in one task, and saccades in another. EMG and eye tracking was used to monitor effector activity. The authors found that, prior to the decision, activity ramped up in a manner consistent with diffusion model accounts. However, they also found that just prior to the decision, activity differed as a function of task difficulty as defined by the temporal density of auditory clicks. An analysis of region-specific data indicated that the parietal region was The authors conclude that the choice process is based on graded rather than threshold mechanism.

This is a well-conducted and well-motivated study with interesting findings and potentially impactful implications. Threshold-based neural signaling has been reported quite frequently, so the current findings may be at odds with many previous findings, or may offer a nuance that has been missed by

single unit and neuroimaging techniques. As the authors note, confidence may have interacted with the strength of evidence. Overall, I am quite enthusiastic about this study but have one major issue that I think should be addressed.

I was surprised that the findings were not contrasted more directly with previous reports, with some discussion of potential explanations. In particular I think it would be important to compare the current findings with those of Hanes and Schall, *Science*, 1996. In that study, the authors measured electrical activity from the motor cortex of monkeys performing a decision task. As with the current study, they evaluated whether neural activity prior to the choice adhered to a threshold or graded model. They found that the data supported a variable drift to common threshold account across decision times. At least in motor cortex. Thus, their data (like most) are at odds with the reported findings. The current study included motor regions and the effector specific analysis of regional contributions indicated that premotor areas were important (along with parietal). I think the authors need to make some attempt to integrate their results with findings such as these.

One minor comment about the effector data is that I have seen some neuroimaging studies speculate about the role of effectors in evidence accumulation and, given the significant relationship between effector activity and DV, I think the manuscript would benefit from a bit more discussion about the possibility while citing those papers. Many accounts assume that evidence is tracked centrally, but it seems important to also consider that people will offload the work when possible. In this case, is it offloaded to peripheral or muscular systems?

Reviewer #4 (Remarks to the Author):

This manuscript presents neural recordings from epilepsy patients, who were implanted with electrodes for presurgical evaluation, while performing a perceptual decision task. Earlier animal work suggested that the conversion of a graded representation of accumulated sensory evidence into a discrete choice could be based on a threshold mechanism, such that the decision process is terminated when the activity of a pool of neurons supporting a particular choice reaches a criterion level. While the authors found ramping signals that appeared decision-related, these signals did not exhibit a stereotyped activity level around the time of choice. The authors conclude that no thresholding mechanism was at work, when their subjects made their decisions.

While many animal experiments have been performed to study the mechanisms of perceptual decision-making, and while human EEG has been recorded during such tasks, the study is novel in the sense that it is certainly one of the first human intracranial recordings during perceptual decision-making that I am aware of. I have some issues with how the authors stated the problem that the manuscript addresses,

and I believe that the data are overinterpreted: Not having observed signals in their recordings that reached a common threshold around the time of choice does not necessarily mean that such signals didn't exist in the brain. I will further elaborate on these points below.

Major issues:

1) When setting up the topic of the manuscript (first page), the authors state "Whether decisions are definitive – concluding in all-or-none choices, or whether the underlying representations are graded, has been unclear." and "[...] our decisions and choices appear to be associated with a degree of certainty. Such introspective evaluations suggest a graded instead of an all-or-none process.". From my point of view, it is the very nature of a decision process that some form of graded representation has to be converted into a discrete choice. I therefore don't see how this could not be the case. The interesting question to ask is HOW this is achieved. Although the decision ends with a discrete choice, the underlying graded representation does provide information about aspects like certainty. This is not expected to be any different for highly trained animals. Kiani & Shadlen (2009), for example, gave monkeys an option to either report a choice with the chance of earning a large reward when making a correct choice or to opt for a small, but certain reward. The monkeys opted for the latter in situations, where humans would report high uncertainty associated with the choice. The claim that a threshold crossing mechanism might be specific to highly trained animals further surprises, as related observations have also been reported in human EEG (e.g., Kelly & O'Connell, 2013).

2) In the animal literature, the claim that neural activity seemed to converge to a common firing rate around the time of choice was based on single-unit recordings from carefully selected neurons with a clear preference for one of the possible choices. The signals that are reported in this manuscript are gamma-band LFP/ECoG recordings, mostly from the cortical surface (7 out of 8 subjects). The signals are expected to be not very localized and therefore correlated with multi-unit activity of larger, likely inhomogeneous populations of neurons. This view is supported by Fig. S5, which indicates that, within the same brain area (area 40 is a good example, but the same is true for others), decision-related signals with very different properties were found: signals that ramped up for both saccade and joystick responses (both possible choices), signals that only ramped up for saccade responses, and signals that only ramped up for joystick responses. Given that the recorded signals are likely mixtures of decision-related signals in the brain, it is not too surprising that the recorded signals did not show a stereotyped activity level around the time of choice. For example, Roitman & Shadlen (2002), when recording from monkey LIP during a visual perceptual decision task, reported a stereotyped firing rate of neurons being part of the winning pool just prior to the animal reporting its choice. Neurons preferring the other choice, however, did not have this property. Their firing rate was still graded around the time of choice. Any mixture of these signals would therefore no longer be expected to show a stereotyped response level around the time of choice. Not being able to find such a property in LFP/ECoG signals, most of which were recorded with surface electrodes, should therefore not be interpreted as indicating that no such signals existed in the brain. The authors even state in the second-to-last paragraph that "The graded gamma activity could represent the DV itself, the confidence in a choice, or any other correlated

variables.” Exactly! And many of these would be expected to be graded, even if there were a threshold crossing mechanism involved at some point during the decision process that was not directly observable in the signals recorded in this study.

3) It is not clear to me whether any of the reported signals are a direct neural correlate of the decision variable (DV). They are certainly decision-related in some way, but a neural signal directly reflecting the DV should be reflecting the accumulated net sensory evidence while subjects experience the sensory stimulus, i.e., the slope of the ramping signal should be reflecting whether there is more evidence for left or more evidence for right and how much. These stimulus-specific ramping signals typically start approx. 100-200 ms after stimulus onset. (Fig. 1 in Hanks et al. (2015) provides an example for rats performing the clicks task.) In this manuscript we only get to see the first 300 ms after stimulus onset (everything else is plotted relative to the onset of the motor response). What happens after the first 300 ms? And does this stimulus-locked response reflect accumulated evidence? If it doesn't, we are not even looking at a neural signal that could be used for making the choice. Related to this point, what would the traces in Fig. 3b look like for the opposite choice?

Minor issues:

4) The probabilistic relationship between perfectly accumulated evidence and choice shown in Fig. 6 is also not a human-specific phenomenon. Highly trained rats show it, too (Brunton et al., 2013), and it is expected assuming that the evidence accumulation process is noisy and/or leaky rather than perfect.

5) “We further corrected the eye movement onset with 33 ms relating to the limited resolution of the eye tracker.” (page 13): I assume the (systematic) correction was applied due to a known processing delay rather than the limited resolution?

6) “The stimuli consisted of a train of brief click sounds drawn from a homogeneous Poisson distribution, which has an equal mean for left/right ears.” (page 12) & “A correct response was indicated by a green text ([...] in the order of increasing stimulus difficulty) [...]” (page 13): How was stimulus difficulty varied, if the Poisson rates for the left and the right ear were always identical? Did this common rate vary across trials?

7) “Electrodes showing significant gamma modulation for both saccades and button press choices were defined as effector-independent task-related electrodes.” (page 15): What about the direction of the modulation? Were there electrodes that showed an upward modulation for one choice and a downward modulation for the other choice? Or were these always congruent (upward)?

8) “Specifically, we generated a modeled saccade (button press) DV with the value linearly ramped from 0 to 1 (-1) [...]” (page 16): Using a model with a combination of upward and downward ramps doesn’t make a lot of sense to me, if none of the recorded signals showed any evidence of downward ramping responses. (I certainly haven’t seen any examples.)

9) In subjects, who performed both the congruent task and the reversed task, were trials just collapsed across both conditions, or analyzed separately? If the former, I could see a potential issue, as the direction of the required saccade reversed across task conditions.

References:

Brunton BW, Botvinick MM, Brody CD (2013) Rats and Humans Can Optimally Accumulate Evidence for Decision-Making. *Science* 340:95-98

Hanks TD, Kopec CD, Brunton BW, Duan CA, Erlich JC, Brody CD (2015) Distinct relationships of parietal and prefrontal cortices to evidence accumulation. *Nature* 520:220-223

Kelly SP, O’Connell RG (2013) Internal and External Influences on the Rate of Sensory Evidence Accumulation in the Human Brain. *J Neurosci* 33:19434-19441

Kiani R, Shadlen MN (2009) Representation of confidence associated with a decision by neurons in the parietal cortex. *Science* 324:759-764

Roitman JD, Shadlen MN (2002) Response of Neurons in the Lateral Intraparietal Area during a Combined Visual Discrimination Reaction Time Task. *J Neurosci* 22:9475-9489

Response to Comments by Reviewers:

We thank the Reviewers for the time and attention they devoted to reviewing this article, for acknowledging the effort that went into performing the invasive human recordings under highly controlled experimental conditions, and for the helpful comments. We have performed the necessary work to address all comments. Major revisions include:

1. A new Introduction that provides a thorough account of existing literature and clarifies and highlights the specific contribution of this article.
2. New analyses that confirm the representation of the decision variable by the gamma neural activity.
3. New analyses, including the inspection of reaction time, which refute models such as secondary post-processing of decisions.
4. Dedicated analyses to answer additional points of the Reviewers, along with the updated text.

The following paragraphs provide our detailed responses to each individual point.

Reviewer #1:

A large body of previous neurophysiological investigations in humans and non-humans have highlighted decision signals that exhibit a threshold-crossing relationship with choice behaviour, reaching a stereotyped amplitude immediately prior to a choice report. The present paper sought to determine whether the brain carries signatures of the evolving decision that are graded, right up to the point of commitment, consistent with a representation of confidence. Intracranial EEG data were collected from 8 pre-surgical patients performing a perceptual discrimination task (poisson clicks) in which saccades and hand button presses were used to report choices and stimulus-response mappings were varied across sessions. Data analyses centered on gamma band activity. The authors first identified sensors showing significant increases in activity at response relative to baseline for both response modalities (termed 'effector-independent') and those that only showed activity increases for one modality (termed 'effector-specific') and then examined the degree to which pre-response activity at these sensors scaled with the estimated DV extracted from the physical stimulus sequence. The results highlight a distributed set of sensors exhibiting a graded relationship with the DV covering primarily parietal, premotor and prefrontal areas.

I found this paper a very interesting read and the data are clearly of very high value. Although there are only 8 patients included, there was excellent sensor coverage of the cortex which is a key strength of the paper. I was also impressed with the authors' efforts to exclude purely sensory-evoked signals by first eliminating any sensors that were activated in response to the clicks during passive listening. However, I do have a number of significant concerns about key aspects of the data analysis methods as well as with the framing of the paper with respect to the prior literature.

We thank the Reviewer for the enthusiasm about the study and the constructive feedback. We address the individual points in detail below.

Major Comments

1. I found the rationale for the paper as framed in the Abstract and Introduction very confusing. It feels to me like a straw man argument to suggest that the field is in the midst of a debate

between those who believe that the brain makes decisions in a purely all or none fashion and those who believe the brain carries graded representations of choice confidence. I have not come across anyone who would hold to the former view. It is true that the standard drift diffusion model assumes a fixed bound at response but numerous alternative models exist that allow for time-varying bounds that are not acknowledged by the authors. For example, there is now strong evidence from neurophysiological investigations and modeling studies supporting the role of collapsing bounds or urgency in decision making (e.g. Hanks et al 2014, eLife; Murphy et al 2016, Nat Comms) whereby the brain progressively lowers the cumulative evidence required to trigger commitment as time passes. Further, relationships between pre-response decision signal amplitudes and cumulative evidence have been reported in non-invasive human brain recordings (e.g. Kelly et al 2021, Nat Hum Beh; Steinemann et al 2018, Nat Comms) and direct recording studies in monkeys have demonstrated a graded relationship between the spike rates of decision variable encoding neurons and choice confidence (e.g. Kiani et al 2009). So with all this in mind it is hard to discern exactly what theory or hypothesis the authors are seeking to test here. It seems to me like the authors would be better off focussing on the goal of localising brain regions that exhibit certain decision-related characteristics given that the human brain's decision making architecture remains quite mysterious and has proven difficult to definitively map through non-invasive means.

We thank the Reviewer for the thoughtful comments. We have addressed each component of this question separately:

Subquestion #1: The bound vs. no bound hypothesis at the time of response.

We agree with the Reviewer that several models could lead to varying bounds, such as collapsing bound and attractor network models. We have summarized these models in the revised Introduction:

"...For instance, humans and animals generally make decisions under time constraints, which exert limits on the available decision time [1, 3, 6, 34–39]. These constraints have been modeled by collapsing bounds that decrease their levels over the decision time [28, 40]. Furthermore, there are alternative, multi-dimensional attractor network models that do not require a definition of a bound [41–44]. In attractor network models, decision-related activity evolves in a multidimensional and hierarchical space involving many brain regions and neurons until it reaches a stable state defined by time constraints, accuracy requirements, and other decision-relevant variables. Both the collapsing and attractor network models allow for decision-related activity to be graded by evidence accumulated at the time of choice (Fig. 1b)."

Although there are several studies that have found graded representation of choice confidence, this study specifically investigated the representation of the decision variable (DV), not the choice confidence (the two are related but not equivalent). With respect to DV (as opposed to choice confidence), there is a substantial debate regarding whether the fixed or variable boundary model is appropriate. In the DV realm, multiple groups hold the view that evidence is accumulated to constant decision boundaries. For example, Ratcliff et al. (2016) suggested that diffusion models with constant decision bounds provide the best account of the choice behavior. Moreover, a recent study by Smith et al. in 2022 found that the standard diffusion model with a constant decision bound was the most appropriate model for tasks with constant stimulus information, and that models with time-varying urgency or decision bounds performed similarly well as the standard diffusion model on tasks with changing stimulus information. In the revised manuscript, we now cite the literature that supports the theory of a fixed bound at the time of response:

“...It has been found that many brain regions track the accumulated evidence for a decision [1–13], which has been captured by a dominant, drift-diffusion model of decision-making [14–16]. However, it has been unclear how the accumulation process concludes. The traditional view posits that choices are made in an all-or-none manner [14–25], when a neural signal that represents a forming decision reaches a defined bound (Fig. 1a).”

In addition to citing the literature in support of a bound, we also cite literature that highlights the debate about whether DV signals reach a bound at the time of a response in the revised Introduction:

“...More recently, this view of the decision process has been criticized. In particular, it has been questioned whether the concept of a decision bound provides an inclusive-enough account of behavioral and neural data [26–33].”

- Ratcliff, R., Smith, P. L., Brown, S. D., & McKoon, G. (2016). Diffusion decision model: Current issues and history. *Trends in Cognitive Sciences*, 20(4), 260–281.
- Smith, P. L., & Ratcliff, R. (2022). Modeling evidence accumulation decision processes using integral equations: Urgency-gating and collapsing boundaries. *Psychological Review*, 129(2), 235–267.

Subquestion #2: The theory or hypothesis the authors are seeking to test in this study needs to be clearly presented.

The study seeks to A) resolve the decision-bound-related debate using direct neural recordings in humans, and B) characterize the brain regions involved in the decision process. This is now captured in the following text in the revised Introduction:

“The two prevalent models (Fig. 1) have been difficult to tease apart. Computational models often make similar predictions of choice behavior even though they invoke fundamentally different neural mechanisms [25, 28]. The models could be distinguished using direct recordings of neural activity, but such recordings have thus far only been conducted in particular brain regions of non-human primates [1–5]. Since specific brain regions encode different aspects of forming decisions, a conclusive answer to how the brain represents the entirety of the decision process in humans has remained elusive. On the other hand, human studies, which have used non-invasive modalities [6–13], could only access broadly distributed low-frequency signals, which provides a coarse perspective on the underlying neural processes.

To address this issue, we have recorded neural activity directly from multiple regions of the brain of humans making perceptual decisions (Fig. 2, Fig. 3). The broad coverage of the intracranial recordings and their high fidelity enabled us to characterize the neural dynamics and the brain regions involved in decision formation.”

Subquestion #3: The study should focus on localizing the brain regions that exhibit decision-related information

We agree that the neural correlates of developing decisions in the human brain have been largely unexplored, given that non-invasive methods cannot provide the necessary granularity. Therefore, identifying the brain regions involved in decision formation has been a main component of the study. This is now stressed in the following text in the revised Introduction:

“To address this issue, we have recorded neural activity directly from multiple regions of the brain of humans making perceptual decisions (Fig. 2, Fig. 3). The broad coverage of the intracranial recordings and their high fidelity enabled us to characterize the neural dynamics and the brain regions involved in decision formation.”

Our invasive recording had broad brain coverage. This allowed us to localize brain regions that show decision-related signals. We found that BA40 (parietal cortex) has the most prominent contribution to the grading of broadband gamma activity with the DV at the time of choice for both saccade and button press choices. Additionally, we found BA40, BA8 (including frontal eye fields) for saccade choice; and BA40, BA6 (premotor/supplementary motor areas) for button press choice have the most prominent contribution to the grading effect. These findings were available in the paper previously but are now stressed in the following sentence:

“...For saccade and button press-modulated regions (Fig. 5a; Fig. S3b), we found that BA40 (parietal cortex) has the most prominent contribution to the grading of the γ activity. For saccade or button press-modulated regions (Fig. 5b; Fig. S3d) we found BA40, BA8 (includes frontal eye fields) for saccade choice; and BA40, BA6 (premotor/supplementary motor areas) for button press choice to have the most prominent contribution to the grading of the γ activity.”

We further highlighted the necessity of exploring decision-related information with invasive recording in humans in a new paragraph in the revised Discussion:

“There is emerging evidence that neural signals underlying perceptual decisions made by humans may not be limited by a bound [6–13]. However, the noninvasive EEG recordings in these studies only provide low-frequency signals that represent a summation of electric potentials across many brain regions. Therefore, the EEG recordings could only provide a coarse perspective on the underlying neural processes. Here, we used invasive recordings and evaluated localized high-frequency γ activity, which is a surrogate of multiunit discharge activity. The invasive recordings provide direct evidence that the human brain can implement decisions in an analog manner.”

2. The introduction also makes much of the fact that signals exhibiting threshold-crossing relationships to behaviour have tended to be observed in experiments with highly-trained participants. The authors note then that a goal of the study was to examine decision signals in a context where participants have not had much exposure to the task. This aspect of the paper is treated in quite a vague way. First, it is not clear what the authors consider ‘highly trained’. Their participants underwent hundreds of trials worth of testing so that would appear to be pretty comparable to most human investigations of decision making. Second, the authors appear to be implying then that the reason they do not see threshold-crossing effects in their data is because the participants had not been extensively trained. This claim is just not supported by the data for two reasons. First, because the authors have not conducted a systematic manipulation of task exposure and,

To address this point, we have eliminated the claims regarding training as a studied factor from the Introduction. The revised Introduction now only focuses on the aspects that were studied and for which there are data. In particular, the Introduction now reads:

“The two prevalent models (Fig. 1) have been difficult to tease apart. Computational models often make similar predictions of choice behavior even though they invoke fundamentally different neural mechanisms [25, 28]. The models could be distinguished using direct recordings of neural activity, but such recordings have thus far only been conducted in particular brain regions of non-human primates [1–5]. Since specific brain regions encode different aspects of forming decisions, a

conclusive answer to how the brain represents the entirety of the decision process in humans has remained elusive. On the other hand, human studies, which have used non-invasive modalities [6–13], could only access broadly distributed low-frequency signals, which provides a coarse perspective on the underlying neural processes.

To address this issue, we have recorded neural activity directly from multiple regions of the brain of humans making perceptual decisions (Fig. 2, Fig. 3). The broad coverage of the intracranial recordings and their high fidelity enabled us to characterize the neural dynamics and the brain regions involved in decision formation.”

second, because the authors have not conducted analyses that could rule out the possibility that threshold-crossing signals may be present in their data. As far as I can see, no analyses are provided to identify sensors that exhibit a threshold-crossing effect (e.g. minimum amplitude variance in the pre-response measurement window) and the focus on searching for sensors that correlate with the DV would exclude such signals by definition.

We acknowledge that it is possible that there are other regions involved in evidence accumulation towards a bound besides those inspected with the implanted electrodes here. In particular, this is now included as a potential shortcoming in the revised Discussion:

“... it is important to stress that only a subset of the recorded regions showed modulation by the DV at the time of choice. It is, therefore, possible that there are other regions involved in evidence accumulation towards a bound besides those inspected here. Implanting electrodes into many more brain regions (including the deep brain regions) would help to address this matter.”

As a more minor point, if the authors were to have strong evidence for the absence of threshold-crossing signals in their recordings, then they should provide some Discussion around the fact that this rules out what has been assumed to be the key stopping mechanism for the brain's decision process. In other words, if the signal identified in this study have a graded relationship with the DV then what determines the point at which commitment is reached and evidence accumulation terminated?

There are attractor and collapsing bound models that allow for graded neural responses while leading to a defined behavioral response. These models render the decision process as a continuum in an attractor space, not necessarily as a binary process in the traditional one-dimensional space. We now introduce these alternative models right in the revised Introduction:

“... For instance, humans and animals generally make decisions under time constraints, which exert limits on the available decision time [1, 3, 6, 34–39]. These constraints have been modeled by collapsing bounds that decrease their levels over the decision time [28, 40]. Furthermore, there are alternative, multi-dimensional attractor network models that do not require a definition of a bound [41–44]. In attractor network models, decision-related activity evolves in a multidimensional and hierarchical space involving many brain regions and neurons until it reaches a stable state defined by time constraints, accuracy requirements, and other decision-relevant variables. Both the collapsing and attractor network models allow for decision-related activity to be graded by evidence accumulated at the time of choice (Fig. 1b).”

Furthermore, in response to this comment, we now provide specific potential reasons that may demand the use of the graded model in the revised Discussion:

“Decision-makers live in dynamic environments with varying goals and behavioral demands. To make effective decisions, individuals must gather sensory evidence and use it to plan actions in a particular context. The context must consider relevant stimulus sensory information, relevant actions

and their payoffs, and the mapping between the stimuli and the actions [33]. This contextual processing demands a flexible representation of the decision process. In this light, the dominant, bounded model (Fig. 1a) applies only to a subset of decisions performed in stationary contexts, in which subjects typically perform hundreds of similar discrimination decisions consecutively, applying the same computations and task rules across trials to approximate optimal decisions. In natural settings, decisions are almost exclusively made in new and dynamic contexts, which include variable decision policies that rest on factors such as urgency, reward expectation, speed-accuracy trade-offs, stimulus-action mapping, and decision confidence [33]. The graded model (Fig. 1b), in which neural activity does not need to reach a defined level in the decision process, can accommodate these factors.”

3. The criteria for classifying sensors as ‘effector-independent’ and ‘effector-selective’ appears problematic. It seems highly likely that motor preparation areas that are selective for hand movements might still exhibit some activity during decisions that were ultimately reported via saccades and vice versa - in fact it would be very surprising if this were not the case. According to the authors' current approach such signals would end up being classed as ‘effector-independent’. The authors do show using eye tracking and EMG that there is no apparent systematic activation of the unused effector but this does not mean that we can assume that there is no preparation of these responses at the level of the cortex and many models and previous neurophysiological observations would predict that there should be some accumulation of evidence for the unchosen alternative. A plot showing gamma waveforms for saccade-selective sensors on trials that were reported with button pushes (and vice versa) would be really important to include. Regardless, it doesn’t seem appropriate to class a sensor as ‘effector-independent’ when it’s possible that a sensor could show significant activation for both response types and yet show significantly greater activation for one response-type than the other.

We have addressed this comment in two ways:

First, as suggested, we now provide a plot showing gamma waveforms for saccade-selective (Fig. S7b) sensors on trials reported with button press and vice versa. We found the BP-selective regions were not graded by the DV at the time of saccade choices, and the SC-selective regions were indeed not graded by the DV at the time of button press choices (Fig. S7b, also shown below). This is additionally reported in the associated text: “...The BP-selective regions were not graded (*ns*, $p > 0.05$) by the DV at the time of saccade choices (top histogram) with an average $R = 0.07$ ($t(12) = 2.0$, $p = 0.07$; two-tailed *t*-test). Similarly, The SC-selective regions were not graded by the DV at the time of button press choices (bottom histogram) with an average $R = -0.0$ ($t(9) = -0.0$, $p = 1.0$; two-tailed *t*-test).”

Figure S7

Second, we eliminated the effector-specific/independent framing and instead provided the reader with specific information about the particular kind of modulation. Specifically, in the revised manuscript, we show “SC&BP-modulated” regions as those that showed significant broadband gamma response for both saccade (SC) and button press (BP) choices (Fig. 3a); and “SC-modulated” or “BP-modulated” regions as those that showed significant broadband gamma response for saccade or button press choices (Fig. 3b). Similarly, we show “SC&BP-graded” regions as those that showed graded effects at the time of choice for both saccade and button press choices (Fig. 5a); and “SC-graded” or “BP-graded” regions as those that showed graded effects at the time of choice for saccade or button press choices (Fig. 5b).

The definition of the “SC&BP-modulated” electrodes is provided in the revised Methods: *“...Electrodes showing ($p < 0.05$) significant difference (increase or decrease) in γ activity at response relative to baseline were identified as effector-modulated electrodes. Electrodes showing significant γ modulation for both saccade (SC) and button press (BP) choices were defined as SC&BP-modulated electrodes (Fig. 3a).”*

The definition of the “SC-modulated” and “BP-modulated” electrodes is provided in the revised Methods: *“...Electrodes showing significant γ modulation for saccade choices were defined as SC-modulated electrodes, and significant γ modulation for button press choices as BP-modulated electrodes (Fig. 3b).”*

The definition of the “SC&BP-graded” electrodes is provided in the revised Methods: *“...Electrodes positively graded by both saccade and button press DV were defined as SC&BP-graded electrodes (Fig. 5b, Fig. S3b).”*

The definition of “SC-graded” and “BP-graded” electrodes is provided in the revised Methods: *“...Electrodes with the γ significant ($p < 0.05$) positively graded by saccade (button press) DV at the time of choice were defined as SC (BP)-graded electrodes (Fig. 5a, Fig. S3d).”*

4. A total of 13 sessions were recorded in the two tasks and the key statistical tests are performed across sessions rather than participants. Given that only 55% of trials were valid, there is a need to establish whether or not certain trial types and participants were more prominent in some of the cells of the analysis. For example, can the authors exclude the possibility that the signal scaling with DV is not due to different mixtures of saccade vs button responses across difficulty? This is a particular concern because there appear to be systematic differences in the slope of the relationship between RT and DV for saccade vs manual responses.

To avoid this potential issue of mixing, throughout the manuscript, we have separated saccade trials and button press trials. This is now stressed in the revised Methods as the following text: *“...Importantly, to avoid the potential mixing of decision difficulty and response type, we have separated saccade and button press trials throughout the manuscript.”*

Similarly can the authors rule out confounds due to imbalances in the number of congruent vs incongruent trials across difficulty levels

In our study, the same principal finding about the grading effect holds across congruent and reversed trials (Fig. S3f). In further response to the reviewer’s comment, we found both congruent and reversed types comprise a balanced number of trials, and this is reported in a new Fig. S1d in the revised manuscript and the associated text: *“...Specifically, subjects*

performed an average of 218 (241) congruent trials for the saccade (button press) choice, while subjects performed an average of 205 (252) incongruent trials...”; “...The analysis of the congruent and reversed data was also performed separately.”

Additionally, subjects performed only one task type (congruent or incongruent) in each session. This is now stressed in the revised Methods as the following sentence. Therefore, the design prevented congruent and incongruent task effects from being mixed. “...We collected $n = 7$ sessions in the congruent task and $n = 6$ sessions in the reversed task. Subjects performed only one task type (congruent or reversed) in each session. Five of the 8 subjects performed both tasks...”; “... importantly, to avoid the potential mixing of decision difficulty and response type, ..., The analysis of the congruent and reversed data was also performed separately.”

Figure S1

or variations in the presence/absence of certain sensors in a given session?

All electrodes involved in the congruent task session analysis were identical to those in the reversed task. This is now stressed in the revised Methods as the following text: “...Electrodes involved in the congruent task session analysis were identical with those in the reversed session.”

Minor Comments:

In the introduction the authors state “This debate has been difficult to resolve at the general level because there has been no study that has i) recorded neural activity at high spatiotemporal coverage directly from the brain”. The authors would need to clarify why direct recordings are necessary to resolve the ‘debate’ given that graded signals have already been identified in non-invasive recording studies.

As mentioned above, the Introduction is now rewritten such that it focuses on the pertinent question. In particular, the revised Introduction now reads:

“The two prevalent models (Fig. 1) have been difficult to tease apart. Computational models often make similar predictions of choice behavior even though they invoke fundamentally different neural mechanisms [25, 28]. The models could be distinguished using direct recordings of neural activity, but such recordings have thus far only been conducted in particular brain regions of non-human primates [1–5]. Since specific brain regions encode different aspects of forming decisions, a conclusive answer to how the brain represents the entirety of the decision process in humans has remained elusive. On the other hand, human studies, which have used non-invasive modalities [6–13], could only access broadly distributed low-frequency signals, which provides a coarse perspective on the underlying neural processes.

To address this issue, we have recorded neural activity directly from multiple regions of the brain of humans making perceptual decisions (Fig. 2, Fig. 3). The broad coverage of the intracranial recordings and their high fidelity enabled us to characterize the neural dynamics and the brain regions involved in decision formation.”

The authors should provide more discussion of the distinct functional roles that the identified effector-independent and effector-selective areas might be playing in the decision process.

To address this point, we have included a new paragraph on *central and embodied cognition* theories in the revised Discussion:

“Many accounts of decision-making assume that decision evidence is tracked in a central cognitive module that is independent of the effector systems that execute the respective choices [60, 61]. Contrary to this view, theories of embodied cognition have proposed that the decision-making process can be offloaded to cognitive faculties and neural circuits that implement the associated choices [62–66]. In our study, we found representations of decision evidence in many brain regions that encode choice, thus contributing to the theories of embodied cognition. Specifically, our data show that decisions can be formed within the same circuits that plan and execute the resulting choice. For example, the graded effect was most prominent over parietal (BA40) and premotor/supplementary motor areas (BA6) for button press choice; and over parietal (BA40) and frontal eye fields (BA8) for saccade choice (Fig. 5). This distributed representation may save cognitive resources and accelerate the production of action [64–66].”

“The graded modulation during both choice kinds (effector-independent, Fig. 5a-top; Fig. S5) was strongest in the parietal cortex (BA 40), while the graded modulation during each choice (effector-specific, Fig. 5b-top; Fig. S5) was most prominent over parietal (BA 40) and premotor/supplementary motor areas” This is not really apparent from the plot it looks like there is a lot of overlap in the distribution of sensors for both signal types

To address this, we tempered the claim to only state that BA40 has the most prominent contribution to the grading of the broadband gamma activity during both choice kinds in the revised Results: “...For saccade and button press (SC&BP-graded, Fig. 5a; Fig. S3b), we found BA40 (parietal cortex) has the most prominent contribution to the grading of the γ activity.”

For the graded modulation during each choice (defined as SC/BP-graded regions in the revised manuscript, Fig. 5b), we found BA40, BA8 (includes frontal eye fields) for saccade choice (Fig. 5b-top); and BA40, BA6 (premotor/supplementary motor areas) for button press choice (Fig.5b-bottom) have the most prominent contribution to the grading of the broadband gamma activity. This is reflected in the revised Results as the following text: “...For saccade or button press-modulated regions (SC/BP-graded, Fig. 5b; Fig. S3d) we found that BA40, BA8 (includes frontal eye fields) for saccade choice; and BA40, BA6 (premotor/supplementary motor areas) for button press choice have the most prominent contribution to the grading of the γ activity.”

Discussion – “Within the traditional drift-diffusion model, a subject could meaningfully set a putative bound (Fig. 1a) only after having learned the statistics of the given decision task. This would require an extensive exposure to the particular task.” I don’t follow the authors’ point here. There is nothing to stop a participant setting a bound on their response right from the very first trial, it’s just that they would require training in order to identify the optimal bound.

To address this, we have rewritten the text in the revised Discussion: “... In this light, the dominant, bounded model (Fig. 1a) likely applies to a subset of decisions that are performed in

stationary contexts, in which subjects make many similar decisions consecutively. In natural settings, decisions are almost exclusively made in new and dynamic contexts, which include variable decision policies that rest on factors such as urgency, reward expectation, speed-accuracy trade-offs, stimulus-action mapping, and decision confidence³³. The graded model (Fig. 1b), in which neural activity does not need to reach a defined level in the decision process, can accommodate these factors. Our task implements these dynamic contextual and task demands by varying the mapping between stimuli and actions (Fig. 2b). Indeed, subjects found the decisions to be challenging (Fig. 2e), with 55.2% of all trials classified as valid under our stringent acceptance criteria. Yet, the subjects were able to make robust evidence-guided decisions, demonstrated by a defined psychometric curve spanning the entire range of choice probabilities (Fig. 2e). In these more general, dynamic decision situations, we have found robust evidence for a graded nature of the decision process (Fig. 1b).”

Reviewer #2:

This is an interesting and somewhat difficult paper that challenges the commonly-held belief that choices at the neural level can be modeled as an evidence accumulation process with a fixed boundary that represents an internal criterion. Using a paradigm previously developed by Kubanek et al (NeuroImage 2013) and high-resolution ECoG recordings, the authors show compelling evidence that the evidence accumulates beyond a decision threshold and that the peak of evidence accumulation is directly correlated to the evidence itself (e.g., clearer signals reach higher boundaries). This implies that not only the relative evidence for different options but the decisions themselves are graded (“analog”) rather than binary and thresholded.

If confirmed, this finding is revolutionary. The idea of evidence accumulation to a threshold, as codified in the vast family of accumulator and drift-diffusion models, has become one of the cornerstones of the cognitive neurosciences, confirmed in multiple experiments in humans (e.g., the work of Forstmann) and nonhuman primates (e.g., Shadlen).

We thank the Reviewer for thoroughly reviewing the study and for recognizing the importance of this topic.

POSITIVES

The manuscript is well written. The data is of high quality, coming from 13 sessions with 8 patients with ECoG recordings. Multiple analyses are provided to show and bolster the points made. The distinction between effector-dependent and effector-independent regions is a particularly solid point, together with the correlation analysis between evidence and gamma activity. In the discussion, the authors address some of the most obvious counterpoints, including the possibility that gamma activity tracks sensory representations rather than choices. The analysis of Figure 3, in which linear ballistic evidence accumulation is regressed out, is particularly elegant and persuasive.

We appreciate the Reviewer's recognition of the approach.

NEGATIVES

The accumulation-to-threshold framework is not online intuitive and appealing but has robust empirical support in the literature finding separate neural correlates of accumulation rates and thresholds. So, it would be interesting to know what the authors could offer as an explanation for

why the graded nature of evidence has not been found before. Some of the original findings, in fact, clearly show evidence accumulation reaching identical thresholds for different trials (most famously, Roitman & Shadlen 2002). These older findings were established with single-cell, spike-train recordings but, as the authors point out, cortical gamma activity is supposed to track these single-cell spikes at an aggregate level. It is hard to compare the results exactly. In Roitman and Shadlen's paper, for example (as well as in Shadlen et al., 2016 Science), firing rates are visualized in response-lock fashion but also divided into classes of response times (300ms, 400ms, and so on), and the accumulation curves, as expected by the theory, are steeper for shorter RTs, where evidence accumulation would have been faster. Because no such analysis is presented, it is hard to draw conclusions. (In fact, I saw no analysis where RTs are provided for the high, medium, and low evidence signals)

We have addressed this comment in two ways:

First, there is, in fact, evidence for the graded effects in the human literature. However, these studies did not make explicit claims about these data, possibly because the signals obtained using noninvasive (EEG) recordings in humans provide a very macroscopic, summed view of the decision process. We have included the following new paragraph in the revised Discussion:

“There is emerging evidence that neural signals underlying perceptual decisions made by humans may not be limited by a bound [6–13]. However, the noninvasive EEG recordings in these studies only provide low-frequency signals that represent a summation of electric potentials across many brain regions. Therefore, the EEG recordings could only provide a coarse perspective on the underlying neural processes. Here, we used invasive recordings and evaluated localized high-frequency γ activity, which is a surrogate of multiunit discharge activity. The invasive recordings provide direct evidence that the human brain can implement decisions in an analog manner.”

Second, we provide specific potential reasons for the graded nature of evidence in a new paragraph in the revised Discussion:

“Decision-makers live in dynamic environments with varying goals and behavioral demands. To make effective decisions, individuals must gather sensory evidence and use it to plan actions in a particular context. The context must consider relevant stimulus sensory information, relevant actions and their payoffs, and the mapping between the stimuli and the actions [33]. This contextual processing demands a flexible representation of the decision process. In this light, the dominant, bounded model (Fig. 1a) likely applies to a subset of decisions that are performed in stationary contexts, in which subjects make many similar decisions consecutively. In natural settings, decisions are almost exclusively made in new and dynamic contexts, which include variable decision policies that rest on factors such as urgency, reward expectation, speed-accuracy tradeoffs, stimulus-action mapping, and decision confidence [33]. The graded model (Fig. 1b), in which neural activity does not need to reach a defined level in the decision process, can accommodate these factors. Our task implements these dynamic contextual and task demands by varying the mapping between stimuli and actions (Fig. 2b). Indeed, subjects found the decisions to be challenging (Fig. 2e), with 55.2% of all trials classified as valid under our stringent acceptance criteria. Yet, the subjects were able to make robust evidence-guided decisions, demonstrated by a defined psychometric curve spanning the entire range of choice probabilities (Fig. 2e). In these more general, dynamic decision situations, we have found robust evidence for a graded nature of the decision process (Fig. 1b).”

Similarly, the diffusion models with a single threshold would require that the accumulation curves for different conditions (high, medium, and low evidence) start at identical points at the

stimulus onset, but then to ramp up at different slopes. This does not seem the case in Fig. 3; however, to judge this, the stimulus-locked and response-locked plots of gamma activity in Figure 3 would need to extend beyond 200ms and or -200ms.

We have addressed each component of this question separately:

Subquestion #1: Do the accumulation curves for different evidence levels start at the same identical point?

We have confirmed that the accumulation curves for different evidence levels (LO, ME, HI) start at an identical point. Specifically, we calculated the correlation between gamma activity and DV as a function of time, and found that this correlation was not significantly different from 0 at the stimulus onset. This is shown in Fig.3c-d (also emphasized by the gray dots).

Figure 3

Subquestion #2: Do the accumulation curves for different evidence levels ramp up at different slopes?

We have performed a new analysis that confirms this. The results are presented in a new Fig. S5a. We have found that the ramping activity is scaled by the DV in both a model-free (raw broadband gamma activity; Fig. S5a, left) and a model-based (linear fitted activity; Fig. S5a, middle) manner. Moreover, this effect was present in each individual trial. To demonstrate that, we have taken the time course of the gamma activity in each trial and correlated that signal with the time course of the DV within the same trial. The resulting distribution of R values (Fig. S5b) shows a highly significant encoding of the trial-wise DV by the gamma activity (the distribution is somewhat broad, as expected, due to the noise associated with such a single-trial analysis). Besides the new figure, this finding is reported in the revised Results as the following text: “...Moreover, we found that the time course of γ activity correlated with the time course of the DV during each decision period (Saccade: $R = 0.05$, $p = 8.4 \times 10^{-21}$, $t(2319) = 9.4$; Button press: $R = -0.06$, $p = 8.1 \times 10^{-40}$, $t(2698) = -13.4$; two-tailed t -tests).”

Figure S5

Subquestion #3: The authors need to extend the stimulus-locked and response-locked plots of gamma activity.

In the revised manuscript, we show the stimulus-locked plots of gamma activity from -100ms to 300ms, and the response-locked plots of gamma activity from -300ms to 200ms (Fig. 3). We also show the stimulus-locked plots of gamma activity from -200ms to 600ms in the new Fig. S5a.

A possible explanation that I do not see ruled out is the following. The paradigm used in this study, although established in the field and in use for over a decade, is somewhat peculiar: The two left/right response options are mapped to different effectors (saccade vs. fingers). Of course, in any 2AFC task, there must be two distinct possible behaviors to indicate the preferred response. However, the most typical paradigms use extremely easy, consistent, and compatible mappings (left/right choice → left/right saccade). Even with these compatible mappings, and especially with animal models, subjects are extensively trained before the experiment. Both compatibility and training reduce the time necessary to translate the internal decision to the corresponding action. If indeed such a decision takes a significant amount of time (let's say, > 100ms), evidence would continue to accumulate in that period even if a decision has been made. In other words: If the mapping requires a significant second processing stage after the decision threshold has been reached, it would seem like the thresholds are different when they are not. A good way to rule this out would be to visualize a larger window of the gamma-activity response-locked plots in Fig. 3, and visualize the trials by RT bins (200-300ms, 300-400ms, 400-500ms, and so on; like in Shadlen et al 2016 Science). If there is a significant second stage before the motor/saccade response, the traces should appear to cross over in that window.

This possibility is ruled out by the data. In particular, Fig. 3c,d show that the correlation between the DV and the gamma *decreases or drops sharply*, rather than increases or stays flat, following a choice. This is especially clear for saccades, where the correlation between

DV and gamma drops sharply following a choice. This argues against the accumulation continuing following a choice.

Figure 3

This observation is highlighted in the following new text in the revised Results:

“... Notably, the two plots show that the evidence accumulation process ceases following a choice. For both saccade and button press choices, there is a marked decrease in the correlation values immediately following a choice. This result argues against potential post-processing that may take place during involved decisions.”

A second possible explanation is that the graded accumulation is a side effect of the nature of the task. Drift diffusion models assume that evidence might fluctuate randomly between two choices before two decisions are made. Virtually every paradigm I can think of (e.g., random dot motion, whiteness contrast...) uses stimuli where the stimulus does not really change over time, the drift is assumed to be due to noisy sampling, and participants are free to respond whenever they have sampled sufficient evidence. However, as far as I understand this specific paradigm, it is possible that clicks accumulate unevenly for the left and right ear (and Figure 2C seems to support that) and the clicks are provided for a fixed 2s period. In this case, participants might be following a very different strategy, waiting for the entire 2-second period before making a decision. In this case, one would naturally expect that different stimuli would be associated with different amounts of evidence, because, in fact, participants would not be using a threshold at all – they would simply wait until the end and pick the location with the highest momentary evidence. To rule out this possibility, it would be important to have plots that showcase the distribution of response times as well as accuracies and gamma activities because response times would give an indication of the strategy used. If the majority of response times are concentrated on or after the 2-sec mark, then it would be evidence that participants are not using an internal threshold as a decision, but are simply waiting for the clicks to end.

This possibility is refuted by the data. In the revised manuscript, we now show the distribution of reaction times, as a new Fig. S6. The distribution of reaction times, also shown below, demonstrates that most decisions were in fact quite fast, on the order of 0.5 s. Only rarely did a decision take more than 1.0 s, and extremely rarely did a decision take the full available 2.0 s. This finding is now communicated to the reader in the revised Results: *“... Subjects made their decisions rapidly (Fig. S6), well within the 2 s limit.”*

Figure S6

Finally, while the authors provide evidence against the single-threshold model, it is not clear what they would suggest as a replacement. At times, the text seems to imply that they think that the probability of a response simply grows with the corresponding gamma activity, without a definite threshold—sort of a threshold-less drift process? Is this the case that the authors are making? If so, I think it would be important to make it clear how they think that a decision is made. Of course, a full model would be outside the scope of the paper, but it would be interesting to read how the authors think about a viable alternative for how decisions happen.

We are now addressing the potential alternatives to the standard model (with fixed bound) directly in the revised Introduction:

“...For instance, humans and animals generally make decisions under time constraints, which exert limits on the available decision time [1, 3, 6, 34–39]. These constraints have been modeled by collapsing bounds that decrease their levels over the decision time [28, 40]. Furthermore, there are alternative, multi-dimensional attractor network models that do not require a definition of a bound [41–44]. In attractor network models, decision-related activity evolves in a multidimensional and hierarchical space involving many brain regions and neurons until it reaches a stable state defined by time constraints, accuracy requirements, and other decision-relevant variables. Both the collapsing and attractor network models allow for decision-related activity to be graded by evidence accumulated at the time of choice (Fig. 1b).”

Furthermore, in response to this comment, we now provide specific potential reasons that may demand the use of the richer, analog model:

“Decision-makers live in dynamic environments with varying goals and behavioral demands. To make effective decisions, individuals must gather sensory evidence and use it to plan actions in a particular context. The context must consider relevant stimulus sensory information, relevant actions and their payoffs, and the mapping between the stimuli and the actions [33]. This contextual processing demands a flexible representation of the decision process. In this light, the dominant, bounded model (Fig. 1a) likely applies to a subset of decisions that are performed in stationary contexts, in which subjects make many similar decisions consecutively. In natural settings, decisions are almost exclusively made in new and dynamic contexts, which include variable decision policies that rest on factors such as urgency, reward expectation, speed-accuracy tradeoffs, stimulus-action mapping, and decision confidence [33]. The graded model (Fig. 1b), in which neural activity does not need to reach a defined level in the decision process, can accommodate these factors. Our task implements these dynamic contextual and task demands by varying the mapping between stimuli and actions (Fig. 2b). Indeed, subjects found the decisions to be challenging (Fig. 2e), with 55.2% of all trials classified as valid under our stringent acceptance

criteria. Yet, the subjects were able to make robust evidence-guided decisions, demonstrated by a defined psychometric curve spanning the entire range of choice probabilities (Fig. 2e). In these more general, dynamic decision situations, we have found robust evidence for a graded nature of the decision process (Fig. 1b)."

MINOR

I think I understand the correlation analysis, but I am puzzled by some statistics—in particular, the passage from “ $R = 0.12$ and $R = -0.12$ during the saccade and button press choices” to “ $t(12) = 3.2$, $p = 7.3 \times 10^{-3}$ ”; I understand that R values are means across many trials, but how is such a large t -value derived from their distribution?

The relatively high value of the t statistic is due to the relatively small standard deviation of the distribution shown in Fig. 3a,b (right). These tight distributions provide high statistical significance associated with the mean effects.

Reviewer #3:

This study examined the temporal dynamics of electrical activity from implanted electrodes in people who performed an auditory perceptual choice task. To test whether activity prior to a decision is based on graded or threshold (boundary) processes, the authors evaluated broadband gamma activity averaged across electrodes in parietal, frontal, premotor, and motor regions. The task was to identify the laterality of auditory clicks presented to either ear. Click density was randomly chosen from a distribution but later assigned to high, medium, and low strength of evidence. Subjects responded using button presses in one task, and saccades in another. EMG and eye tracking was used to monitor effector activity. The authors found that, prior to the decision, activity ramped up in a manner consistent with diffusion model accounts. However, they also found that just prior to the decision, activity differed as a function of task difficulty as defined by the temporal density of auditory clicks. An analysis of region-specific data indicated that the parietal region was The authors conclude that the choice process is based on graded rather than threshold mechanism.

This is a well-conducted and well-motivated study with interesting findings and potentially impactful implications. Threshold-based neural signaling has been reported quite frequently, so the current findings may be at odds with many previous findings, or may offer a nuance that has been missed by single unit and neuroimaging techniques. As the authors note, confidence may have interacted with the strength of evidence. Overall, I am quite enthusiastic about this study but have one major issue that I think should be addressed.

We thank the Reviewer for the time and attention in reviewing the study and for the enthusiasm about the findings.

I was surprised that the findings were not contrasted more directly with previous reports, with some discussion of potential explanations. In particular I think it would be important to compare the current findings with those of Hanes and Schall, Science, 1996. In that study, the authors measured electrical activity from the motor cortex of monkeys performing a decision task. As with the current study, they evaluated whether neural activity prior to the choice adhered to a threshold or graded model. They found that the data supported a variable drift to common

threshold account across decision times. At least in motor cortex. Thus, their data (like most) are at odds with the reported findings. The current study included motor regions and the effector specific analysis of regional contributions indicated that premotor areas were important (along with parietal). I think the authors need to make some attempt to integrate their results with findings such as these.

We have addressed this comment in three ways:

First, we provide specific potential reasons for the bounded and graded nature of evidence in the revised Discussion:

“Decision-makers live in dynamic environments with varying goals and behavioral demands. To make effective decisions, individuals must gather sensory evidence and use it to plan actions in a particular context. The context must consider relevant stimulus sensory information, relevant actions and their payoffs, and the mapping between the stimuli and the actions [33]. This contextual processing demands a flexible representation of the decision process. In this light, the dominant, bounded model (Fig. 1a) likely applies to a subset of decisions that are performed in stationary contexts, in which subjects make many similar decisions consecutively. In natural settings, decisions are almost exclusively made in new and dynamic contexts, which include variable decision policies that rest on factors such as urgency, reward expectation, speed-accuracy tradeoffs, stimulus-action mapping, and decision confidence [33]. The graded model (Fig. 1b), in which neural activity does not need to reach a defined level in the decision process, can accommodate these factors. Our task implements these dynamic contextual and task demands by varying the mapping between stimuli and actions (Fig. 2b). Indeed, subjects found the decisions to be challenging (Fig. 2e), with 55.2% of all trials classified as valid under our stringent acceptance criteria. Yet, the subjects were able to make robust evidence-guided decisions, demonstrated by a defined psychometric curve spanning the entire range of choice probabilities (Fig. 2e). In these more general, dynamic decision situations, we have found robust evidence for a graded nature of the decision process (Fig. 1b).”

Second, we acknowledge that it is possible that there are other regions involved in evidence accumulation towards a bound (e.g., as shown in the work in Hanes & Schall, 1996) besides those inspected with the implanted electrodes here. In particular, this is now included as a potential shortcoming in the revised Discussion:

“... it is important to stress that only a subset of the recorded regions showed modulation by the DV at the time of choice. It is, therefore, possible that there are other regions involved in evidence accumulation towards a bound besides those inspected here. Implanting the electrodes into many more brain regions (including the deep brain regions) would help to address this matter.”

Finally, there is, in fact, evidence for the graded effects, in the human literature (references [6-13]). However, the authors of those studies did not make explicit claims about these data, possibly because the signals obtained using noninvasive (EEG) recordings in humans provide a very macroscopic, summed view of the decision process. Nonetheless, these findings are now highlighted in a new paragraph in the revised Discussion:

“There is emerging evidence that neural signals underlying perceptual decisions made by humans may not be limited by a bound [6–13]. However, the noninvasive EEG recordings in these studies only provide low-frequency signals that represent a summation of electric potentials across many brain regions. Therefore, the EEG recordings could only provide a coarse perspective on the underlying neural processes. Here, we used invasive recordings and evaluated localized high-frequency γ activity, which is a surrogate of multiunit discharge activity. The invasive recordings provide direct evidence that the human brain can implement decisions in an analog manner.”

One minor comment about the effector data is that I have seen some neuroimaging studies speculate about the role of effectors in evidence accumulation and, given the significant relationship between effector activity and DV, I think the manuscript would benefit from a bit more discussion about the possibility while citing those papers. Many accounts assume that evidence is tracked centrally, but it seems important to also consider that people will offload the work when possible. In this case, is it offloaded to peripheral or muscular systems?

Indeed, we show that decisions can be formed within the same circuits that plan and execute the resulting choice. For example, we found that the graded effect was most prominent in BA40 and BA8 (which includes frontal eye fields) for saccade-graded regions, and in BA40 and BA6 (premotor/supplementary motor areas) for button press-graded regions. Thus, our results support the set of theories of *embodied cognition*, in which the decision-making process can be offloaded to cognitive faculties and neural circuits that implement the associated choices. This is now discussed in the revised Discussion:

“Many accounts of decision-making assume that decision evidence is tracked in a central cognitive module that is independent of the effector systems that execute the respective choices [60, 61]. Contrary to this view, theories of embodied cognition have proposed that the decision-making process can be offloaded to cognitive faculties and neural circuits that implement the associated choices [62–66]. In our study, we found representations of decision evidence in many brain regions that encode choice, thus contributing to the theories of embodied cognition. Specifically, our data show that decisions can be formed within the same circuits that plan and execute the resulting choice. For example, the graded effect was most prominent over parietal (BA40) and premotor/supplementary motor areas (BA6) for button press choice; and over parietal (BA40) and frontal eye fields (BA8) for saccade choice (Fig. 5). This distributed representation may save cognitive resources and accelerate the production of action [64–66].”

Reviewer #4:

This manuscript presents neural recordings from epilepsy patients, who were implanted with electrodes for presurgical evaluation, while performing a perceptual decision task. Earlier animal work suggested that the conversion of a graded representation of accumulated sensory evidence into a discrete choice could be based on a threshold mechanism, such that the decision process is terminated when the activity of a pool of neurons supporting a particular choice reaches a criterion level. While the authors found ramping signals that appeared decision-related, these signals did not exhibit a stereotyped activity level around the time of choice. The authors conclude that no thresholding mechanism was at work, when their subjects made their decisions.

While many animal experiments have been performed to study the mechanisms of perceptual decision-making, and while human EEG has been recorded during such tasks, the study is novel in the sense that it is certainly one of the first human intracranial recordings during perceptual decision-making that I am aware of.

We appreciate the Reviewer recognizing the effort that went into collecting these invasive recordings in humans and the novelty of the study.

I have some issues with how the authors stated the problem that the manuscript addresses, and I believe that the data are overinterpreted: Not having observed signals in their recordings that reached a common threshold around the time of choice does not necessarily mean that such signals didn't exist in the brain. I will further elaborate on these points below.

This constructive feedback is addressed in detail below.

Major issues:

1) When setting up the topic of the manuscript (first page), the authors state “Whether decisions are definitive – concluding in all-or-none choices, or whether the underlying representations are graded, has been unclear.” and “[...] our decisions and choices appear to be associated with a degree of certainty. Such introspective evaluations suggest a graded instead of an all-or-none process.”. From my point of view, it is the very nature of a decision process that some form of graded representation has to be converted into a discrete choice. I therefore don't see how this could not be the case. The interesting question to ask is HOW this is achieved. Although the decision ends with a discrete choice, the underlying graded representation does provide information about aspects like certainty. This is not expected to be any different for highly trained animals. Kiani & Shadlen (2009), for example, gave monkeys an option to either report a choice with the chance of earning a large reward when making a correct choice or to opt for a small, but certain reward. The monkeys opted for the latter in situations, where humans would report high uncertainty associated with the choice. The claim that a threshold crossing mechanism might be specific to highly trained animals further surprises, as related observations have also been reported in human EEG (e.g., Kelly & O'Connell, 2013).

We have removed claims about the training as a factor. Instead, the revised Introduction focuses on the HOW question, i.e., how the neural signals underlying the forming decision:

“The two prevalent models (Fig. 1) have been difficult to tease apart. Computational models often make similar predictions of choice behavior even though they invoke fundamentally different neural mechanisms [25, 28]. The models could be distinguished using direct recordings of neural activity, but such recordings have thus far only been conducted in particular brain regions of non-human primates [1–5]. Since specific brain regions encode different aspects of forming decisions, a conclusive answer to how the brain represents the entirety of the decision process in humans has remained elusive. On the other hand, human studies, which have used non-invasive modalities [6–13], could only access broadly distributed low-frequency signals, which provides a coarse perspective on the underlying neural processes.

To address this issue, we have recorded neural activity directly from multiple regions of the brain of humans making perceptual decisions (Fig. 2, Fig. 3). The broad coverage of the intracranial recordings and their high fidelity enabled us to characterize the neural dynamics and the brain regions involved in the decision formation.”

In addition, we now provide more context and possible reasons for the graded effects.

First, we have introduced the potential alternative models for the graded effects in the revised Introduction, including collapsing bounds and attractor networks:

“...For instance, humans and animals generally make decisions under time constraints, which exert limits on the available decision time [1, 3, 6, 34–39]. These constraints have been modeled by collapsing bounds that decrease their levels over the decision time [28, 40]. Furthermore, there are alternative, multi-dimensional attractor network models that do not require a definition of a bound [41–44]. In attractor network models, decision-related activity evolves in a multidimensional and hierarchical space involving many brain regions and neurons until it reaches a stable state defined

by time constraints, accuracy requirements, and other decision-relevant variables. Both the collapsing and attractor network models allow for decision-related activity to be graded by evidence accumulated at the time of choice (Fig. 1b)."

Second, we now provide specific potential reasons that support the richer, analog model in the following paragraph of the revised Discussion:

"Decision-makers live in dynamic environments with varying goals and behavioral demands. To make effective decisions, individuals must gather sensory evidence and use it to plan actions in a particular context. The context must consider relevant stimulus sensory information, relevant actions and their payoffs, and the mapping between the stimuli and the actions [33]. This contextual processing demands a flexible representation of the decision process. In this light, the dominant, bounded model (Fig. 1a) likely applies to a subset of decisions that are performed in stationary contexts, in which subjects make many similar decisions consecutively. In natural settings, decisions are almost exclusively made in new and dynamic contexts, which include variable decision policies that rest on factors such as urgency, reward expectation, speed-accuracy tradeoffs, stimulus-action mapping, and decision confidence [33]. The graded model (Fig. 1b), in which neural activity does not need to reach a defined level in the decision process, can accommodate these factors. Our task implements these dynamic contextual and task demands by varying the mapping between stimuli and actions (Fig. 2b). Indeed, subjects found the decisions to be challenging (Fig. 2e), with 55.2% of all trials classified as valid under our stringent acceptance criteria. Yet, the subjects were able to make robust evidence-guided decisions, demonstrated by a defined psychometric curve spanning the entire range of choice probabilities (Fig. 2e). In these more general, dynamic decision situations, we have found robust evidence for a graded nature of the decision process (Fig. 1b)."

Finally, there is evidence for the graded effects in the human literature (references [6-13]). However, the authors of those studies did not make explicit claims about these data, possibly because the signals obtained using noninvasive (EEG) recordings in humans provide a very macroscopic, summed view of the decision process. Nonetheless, these findings are now highlighted in a new paragraph in the revised Discussion:

"There is emerging evidence that neural signals underlying perceptual decisions made by humans may not be limited by a bound [6–13]. However, the noninvasive EEG recordings in these studies only provide low-frequency signals that represent a summation of electric potentials across many brain regions. Therefore, the EEG recordings could only provide a coarse perspective on the underlying neural processes. Here, we used invasive recordings and evaluated localized high-frequency γ activity, which is a surrogate of multiunit discharge activity. The invasive recordings provide direct evidence that the human brain can implement decisions in an analog manner."

2) In the animal literature, the claim that neural activity seemed to converge to a common firing rate around the time of choice was based on single-unit recordings from carefully selected neurons with a clear preference for one of the possible choices. The signals that are reported in this manuscript are gamma-band LFP/ECoG recordings, mostly from the cortical surface (7 out of 8 subjects). The signals are expected to be not very localized and therefore correlated with multi-unit activity of larger, likely inhomogeneous populations of neurons. This view is supported by Fig. S5, which indicates that, within the same brain area (area 40 is a good example, but the same is true for others), decision-related signals with very different properties were found: signals that ramped up for both saccade and joystick responses (both possible choices), signals that only ramped up for saccade responses, and signals that only ramped up for joystick responses. Given that the recorded signals are likely mixtures of decision-related signals in the brain, it is not too surprising that the recorded signals did not show a stereotyped activity level around the time of choice. For example, Roitman & Shadlen (2002), when recording from

monkey LIP during a visual perceptual decision task, reported a stereotyped firing rate of neurons being part of the winning pool just prior to the animal reporting its choice. Neurons preferring the other choice, however, did not have this property. Their firing rate was still graded around the time of choice. Any mixture of these signals would therefore no longer be expected to show a stereotyped response level around the time of choice. Not being able to find such a property in LFP/ECOG signals, most of which were recorded with surface electrodes, should therefore not be interpreted as indicating that no such signals existed in the brain. The authors even state in the second-to-last paragraph that “The graded gamma activity could represent the DV itself, the confidence in a choice, or any other correlated variables.”. Exactly! And many of these would be expected to be graded, even if there were a threshold crossing mechanism involved at some point during the decision process that was not directly observable in the signals recorded in this study.

This is indeed a possibility that is nonetheless difficult to answer fully until recordings from the entirety of the brain are available. This is acknowledged as a shortcoming in the revised Discussion:

“... it is important to stress that only a subset of the recorded regions showed modulation by the DV at the time of choice. It is, therefore, possible that there are other regions involved in evidence accumulation towards a bound besides those inspected here. Implanting the electrodes into many more brain regions (including the deep brain regions) would help to address this matter.”

3) It is not clear to me whether any of the reported signals are a direct neural correlate of the decision variable (DV). They are certainly decision-related in some way, but a neural signal directly reflecting the DV should be reflecting the accumulated net sensory evidence while subjects experience the sensory stimulus, i.e., the slope of the ramping signal should be reflecting whether there is more evidence for left or more evidence for right and how much. These stimulus-specific ramping signals typically start approx. 100-200 ms after stimulus onset. (Fig. 1 in Hanks et al. (2015) provides an example for rats performing the clicks task.) In this manuscript we only get to see the first 300 ms after stimulus onset (everything else is plotted relative to the onset of the motor response). What happens after the first 300 ms? And does this stimulus-locked response reflect accumulated evidence? If it doesn't, we are not even looking at a neural signal that could be used for making the choice.

To address this comment, we have performed a new analysis that confirms the encoding of the DV. The results are presented in a new Fig. S5a, also shown below. We have found that the ramping activity is scaled by the DV in both a model-free (raw broadband gamma activity; Fig. S5a, left) and a model-based (linear fitted activity; Fig. S5a, middle) manner. Moreover, this effect was present in each individual trial. To demonstrate that, we have taken the time course of the gamma activity in each trial and correlated that signal with the time course of the DV within the same trial. The resulting distribution of R values (Fig. S5b) shows a highly significant encoding of the trial-wise DV by the gamma activity (the distribution is somewhat broad, as expected, due to the noise associated with such a single-trial analysis). Besides the new figure, this finding is reported in the revised Results: *“...Moreover, we found that the time course of γ activity correlated with the time course of the DV during each decision period (Saccade: $R = 0.05$, $p = 8.4 \times 10^{-21}$, $t(2319) = 9.4$; Button press: $R = -0.06$, $p = 8.1 \times 10^{-40}$, $t(2698) = -13.4$; two-tailed t -tests).”*

Related to this point, what would the traces in Fig. 3b look like for the opposite choice?

We have performed this analysis (new Figure S7b). As expected, there is no significant modulation for the opposite choice. If anything, the signal associated with the opposite choice shows a general decrease prior to a choice.

Minor issues:

4) The probabilistic relationship between perfectly accumulated evidence and choice shown in Fig. 6 is also not a human-specific phenomenon. Highly trained rats show it, too (Brunton et al., 2013), and it is expected assuming that the evidence accumulation process is noisy and/or leaky rather than perfect.

In response to this comment, we have removed the speculation on training being a factor.

5) “We further corrected the eye movement onset with 33 ms relating to the limited resolution of the eye tracker.” (page 13): I assume the (systematic) correction was applied due to a known processing delay rather than the limited resolution?

We have corrected the eye movement onset, which was found to be caused by a processing delay of the eye tracker (Tobii T60, Tobii Technology). We have updated the manuscript accordingly in the revised Methods: “...we corrected the eye movement onset by a measured 33 ms latency of the Tobii T60 eye tracker.”

6) “The stimuli consisted of a train of brief click sounds drawn from a homogeneous Poisson distribution, which has an equal mean for left/right ears.” (page 12) & “A correct response was indicated by a green text ([...] in the order of increasing stimulus difficulty) [...]” (page 13): How

was stimulus difficulty varied, if the Poisson rates for the left and the right ear were always identical? Did this common rate vary across trials?

We thank the Reviewer for pointing this out. The click rates were different for each side. This is now elucidated in the following text in the revised Methods:

“... The stereo stimulus was composed such that the sum of clicks presented to the left ear (Cl) plus the sum of clicks presented to the right ear (Cr) summed to a fixed number $Cl + Cr = \Omega$ in 2 s, $\Omega \in \{50, 64, 78, 92\}$. The value of Ω was drawn randomly on each trial. ...”

7) “Electrodes showing significant gamma modulation for both saccades and button press choices were defined as effector-independent task-related electrodes.” (page 15): What about the direction of the modulation? Were there electrodes that showed an upward modulation for one choice and a downward modulation for the other choice? Or were these always congruent (upward)?

For approximately $\frac{3}{4}$ of the electrodes, we found an upward modulation, while for the remaining $\frac{1}{4}$ of electrodes, we found a downward modulation. We did not distinguish the upward and downward modulation electrodes in our analysis to prevent biased selection. This is now reported in the revised Methods:

“...Electrodes showing ($p < 0.05$) significant difference (increase or decrease) in γ activity at response relative to baseline were identified as effector-modulated electrodes. Specifically, among the effector-modulated regions, we found that $(72 \pm 24)\%$ and $(73 \pm 18)\%$ of electrodes showed significantly increased γ activity for saccade and button press choices, respectively. The remaining effector-modulated electrodes showed significantly decreased γ activity. We included all effector-modulated electrodes in the analysis to prevent biased selection.”

8) “Specifically, we generated a modeled saccade (button press) DV with the value linearly ramped from 0 to 1 (-1) [...]” (page 16): Using a model with a combination of upward and downward ramps doesn’t make a lot of sense to me, if none of the recorded signals showed any evidence of downward ramping responses. (I certainly haven’t seen any examples.)

Indeed, a combination of upward and downward ramps would be problematic had we merged the saccade and button press trials prior to performing the regression analysis. For this reason, we performed the regression between the broadband gamma and modeled DV for saccade and button press choices *separately* (we did not mix the saccade and button press trials). Thus, the definition of negative or positive DV value would not affect the finding of the graded effect. This is now elucidated in the following new text in the revised Methods:

“... Importantly, to avoid the potential mixing of decision difficulty and response type, we have separated saccade and button press trials throughout the manuscript.”

9) In subjects, who performed both the congruent task and the reversed task, were trials just collapsed across both conditions, or analyzed separately? If the former, I could see a potential issue, as the direction of the required saccade reversed across task conditions.

We have performed the analysis in congruent task sessions and reversed sessions separately to avoid the described issue. This fact is now emphasized in the following text in the revised Methods:

“... The analysis of the congruent and reversed data was also performed separately.”

REVIEWERS' COMMENTS

Reviewer #1 (Remarks to the Author):

The authors have clearly put a lot of work into the revisions and the manuscript has been strengthened through the addition of a number of new analyses and clarifications. With respect to the concerns that I shared with the other reviewers about the framing of the paper and the interpretation of the results however I feel the authors have not fully addressed the problem. The authors spend large portions of the manuscript emphasising or implying how their results provide evidence against the threshold-crossing account of decision making while adding a few sentences toward the end of the Discussion flagging that in fact their analyses are not comprehensive and cannot exclude the possibility that such signals reside elsewhere in the brain.

In so doing, the authors appear to be insisting on downplaying previous highly relevant work. First, is the point that several reviewers have raised – that monkey recording data has demonstrated threshold-crossing signals in several effector-selective brain areas. I still do not really grasp the authors' explanation for these previous results in light of their own. I would have expected much more discussion regarding the extent to which we can compare single-unit spiking to LFP and LFP to EEG as this is critical to the authors' conclusions. When I noted that the authors have not actually provided any empirical tests to rule out the presence of threshold-crossing signals, they responded by saying that they had added a section of caveats acknowledging this point. And yet the opening of the Discussion makes the strong claim that 'we found that the broadband gamma activity remained graded at the time of choice, suggesting a graded instead of a definitive decision process.' It is still my view then that the authors would be better off implying that their data being inconsistent with this previous work. The authors' results are interesting in their own right and I think it does the paper a disservice to over extrapolate like this. To my mind, the paper provides a really useful mapping of brain areas exhibiting graded representations of the DV but this need not necessarily exclude the possibility of other functionally distinct representations with threshold-crossing effects.

In the previous round of comments it was flagged that many non-invasive EEG studies have reported on evidence accumulation signals that reach varying amplitudes at choice. These data are essentially dismissed by the authors in their response to reviewer comments and in the manuscript itself because they provide only a 'coarse perspective'. Here the authors appear to be holding these EEG data to a much higher standard than they are willing to hold their own data. As the authors themselves point out, their own recordings are not comprehensive and could overlook threshold-crossing signals. In addition, the LFP data would capture the activity of multiple neurons that may have distinct selectivities and, for all we know, some may cross a threshold and others not. Why are the EEG data to be ignored? Regarding these past EEG studies, the authors state 'However, the authors of those studies did not make explicit claims about these data, possibly because the signals obtained using noninvasive (EEG) recordings in humans provide a very macroscopic, summed view of the decision process.' It is not at all clear what

'explicit claims' the authors are referring to but in actual fact, the major focus of several of these previous studies (e.g. Steinemann et al 2018 Nat Comms; Kelly et al 2021, Nat Hum Beh) was to demonstrate that these signals A) have a graded relationship with RT and choice accuracy that is highly consistent with collapsing bound models and B) evolve in parallel with effector-selective signals that cross a threshold at response.

So overall I think the manuscript would be greatly strengthened if the authors engaged with this previous work to a greater extent. I believe that all of the above can be addressed with some relatively easy changes to the text without undermining the overall impact of the paper in the field.

Reviewer #2 (Remarks to the Author):

I have reviewed all the changes made by the authors, and, as far as I can see, they have addressed all my points.

I appreciate how the authors took the time to walk through their argument and reasoning against some of the points I had raised, in particular about the possible alternative decision strategy I had outlined in my first review. After reading their arguments and looking at their data, I now concur with the authors.

The new figures and analysis address my main concerns. They also address Reviewer 1's concern about the effector dependent/independent ROI assumption, which had escaped me in my first reading of the paper.

The new introduction addresses a much broader set of previous findings, and also addresses more clearly the nature of the underlying problem (the nature of the boundary vs. evidence accumulation).

My only remaining, and minimal, comment: Would the authors consider revising the title to make it more focused? I am not sure that, in its current form, it captures the importance of the findings (this is, of course, an optional recommendation).

Reviewer #3 (Remarks to the Author):

This is a second review. The authors have addressed my questions and, while I am not very convinced by their take on the influence of task type on whether decisions are graded or bounded, I don't have a competing explanation. Though it does make sense that time pressure is better addressed using a graded mechanism. I'm not sure a clear answer is needed here, and am satisfied that they have raised it for discussion (and expanded the introduction). I remain enthusiastic about this study.

Response to Comments by Reviewers:

Reviewer #1:

The authors have clearly put a lot of work into the revisions and the manuscript has been strengthened through the addition of a number of new analyses and clarifications. With respect to the concerns that I shared with the other reviewers about the framing of the paper and the interpretation of the results however I feel the authors have not fully addressed the problem. The authors spend large portions of the manuscript emphasizing or implying how their results provide evidence against the threshold-crossing account of decision making while adding a few sentences toward the end of the Discussion flagging that in fact their analyses are not comprehensive and cannot exclude the possibility that such signals reside elsewhere in the brain.

In so doing, the authors appear to be insisting on downplaying previous highly relevant work. First, is the point that several reviewers have raised – that monkey recording data has demonstrated threshold-crossing signals in several effector-selective brain areas. I still do not really grasp the authors' explanation for these previous results in light of their own. I would have expected much more discussion regarding the extent to which we can compare single-unit spiking to LFP and LFP to EEG as this is critical to the authors' conclusions.

To address this point, we have provided a detailed comparison of distinct recording modalities (i.e., EEG, LFP, and single-neuron recordings) used in decision-making studies within the following new Discussion paragraph:

“...EEG recordings in humans capture low-frequency signals representing a summation of electric potentials across many brain regions. Consequently, EEG recordings offer a limited spatial resolution, providing a generalized perspective on the underlying neural processes. On the other hand, single-neuron recordings in animals have been restricted to specific brain regions, thus hindering the ability to provide a conclusive answer to how the brain represents the entirety of the decision process. Here, we recorded local field potentials (LFPs) directly from the human brain and evaluated localized high-frequency γ activity, which is a surrogate of multi-unit discharge activity [47, 48]. These neural signals provided direct evidence across multiple brain regions that the human brain can implement decisions in a graded manner. Even though multi-unit activity and LFPs are tightly correlated [47, 48], and the standard model states that LFPs/EEG are the extracellular currents primarily reflecting summed postsynaptic potentials of pyramidal cells [71], the empirical literature linking EEG, LFP, and microcircuit neural dynamics is under-explored [72–74]. Therefore, the findings of our study should be interpreted explicitly within localized, high-frequency neural discharges, and under the assumption that neuronal discharges constitute the primary code of decision-related neuronal signaling.”

When I noted that the authors have not actually provided any empirical tests to rule out the presence of threshold-crossing signals, they responded by saying that they had added a section of caveats acknowledging this point. And yet the opening of the Discussion makes the strong claim that ‘we found that the broadband gamma activity remained graded at the time of choice, suggesting a graded instead of a definitive decision process.’ It is still my view then that the authors would be better off implying that their data being inconsistent with this previous work. The authors’ results are interesting in their own right and I think it does the paper a disservice to over extrapolate like this. To my mind, the paper provides a really useful mapping of brain areas exhibiting graded representations of the DV but this need not necessarily exclude the possibility of other functionally distinct representations with threshold-crossing effects.

To avoid over-extrapolation, we have stressed in the following text in the revised Discussion:

“...Even though multi-unit activity and LFPs are tightly correlated [47, 48], and the standard model states that LFPs/EEG are the extracellular currents primarily reflecting summed postsynaptic potentials of pyramidal cells [71], the empirical literature linking EEG, LFP, and microcircuit neural dynamics is under-explored [72–74]. Therefore, the findings of our study should be interpreted explicitly within localized, high-frequency neural discharges, and under the assumption that neuronal discharges constitute the primary code of decision-related neuronal signaling.”

In the previous round of comments it was flagged that many non-invasive EEG studies have reported on evidence accumulation signals that reach varying amplitudes at choice. These data are essentially dismissed by the authors in their response to reviewer comments and in the manuscript itself because they provide only a ‘coarse perspective’. Here the authors appear to be holding these EEG data to a much higher standard than they are willing to hold their own data. As the authors themselves point out, their own recordings are not comprehensive and could overlook threshold-crossing signals. In addition, the LFP data would capture the activity of multiple neurons that may have distinct selectivities and, for all we know, some may cross a threshold and others not. Why are the EEG data to be ignored? Regarding these past EEG studies, the authors state ‘However, the authors of those studies did not make explicit claims about these data, possibly because the signals obtained using noninvasive (EEG) recordings in humans provide a very macroscopic, summed view of the decision process.’ It is not at all clear what ‘explicit claims’ the authors are referring to but in actual fact, the major focus of several of these previous studies (e.g. Steinemann et al 2018 Nat Comms; Kelly et al 2021, Nat Hum Beh) was to demonstrate that these signals A) have a graded relationship with RT and choice accuracy that is highly consistent with collapsing bound models and B) evolve in parallel with effector-selective signals that cross a threshold at response. So overall I think the manuscript would be greatly strengthened if the authors engaged with this previous work to a greater extent. I believe that all of the above can be addressed with some relatively easy changes to the text without undermining the overall impact of the paper in the field.

We have addressed this comment in two ways:

First, as suggested, we have summarized the previous EEG studies (including Steinemann et. al. 2018, Kelly et al 2021, etc.) that indicated the graded effect in decisions in the revised Discussion:

“There is emerging evidence that neural signals underlying perceptual decisions may not be limited by a bound. This evidence comprises electroencephalographic (EEG) recordings in humans [6–13] and single-neuron recordings in animals [1–5] ...”

Second, we provide a detailed description of the differences between EEG and LFP in the revised Discussion:

“...EEG recordings in humans capture low-frequency signals representing a summation of electric potentials across many brain regions. Consequently, EEG recordings offer a limited spatial resolution, providing a generalized perspective on the underlying neural processes. ... Here, we recorded local field potentials directly from the human brain and evaluated localized high-frequency γ activity, which is a surrogate of multi-unit discharge activity [47, 48]. These neural signals provided direct evidence across multiple brain regions that the human brain can implement decisions in a graded manner...”

Reviewer #2:

I have reviewed all the changes made by the authors, and, as far as I can see, they have addressed all my points. I appreciate how the authors took the time to walk through their argument and reasoning against some of the points I had raised, in particular about the possible alternative decision strategy I had outlined in my first review. After reading their arguments and looking at their data, I now concur with the authors.

The new figures and analysis address my main concerns. They also address Reviewer 1's concern about the effector dependent/independent ROI assumption, which had escaped me in my first reading of the paper. The new introduction addresses a much broader set of previous findings, and also addresses more clearly the nature of the underlying problem (the nature of the boundary vs. evidence accumulation).

My only remaining, and minimal, comment: Would the authors consider revising the title to make it more focused? I am not sure that, in its current form, it captures the importance of the findings (this is, of course, an optional recommendation).

To address this comment, we have revised the title to “Graded decisions in the human brain”. This title directly and succinctly reflects the scope of the article.